# Heterogeneity in respiratory electron transfer and adaptive iron utilization in a bacterial biofilm

Yuxuan Qin [1,2], Yinghao He [2], Qianxuan She [2], Philip Larese-Casanova[3], Pinglan Li[1] & Yunrong Chai[2]

In *Bacillus subtilis*, robust biofilm formation requires large quantities of ferric iron. Here we show that this process requires preferential production of a siderophore precursor, 2,3-dihydroxybenzoate, instead of the siderophore bacillibactin. A large proportion of iron is associated extracellularly with the biofilm matrix. The biofilms are conductive, with extracellular iron potentially acting as electron acceptor. A relatively small proportion of ferric iron is internalized and boosts production of iron-containing enzymes involved in respiratory electron transfer and establishing strong membrane potential, which is key to biofilm matrix production. Our study highlights metabolic diversity and versatile energy generation strategies within *B. subtilis* biofilms.

---

[1] Beijing Advanced Innovation Center for Food Nutrition and Human Health, Key Laboratory of Functional Dairy, College of Food Science and Nutritional Engineering, China Agricultural University, Beijing 100083, China. [2] Department of Biology, Northeastern University, Boston, MA 02115, USA. [3] Department of Civil and Environmental Engineering, Northeastern University, Boston, MA 02115, USA. Correspondence and requests for materials should be addressed to P.L. (email: lipinglan@cau.edu.cn) or to Y.C. (email: y.chai@northeastern.edu)

Most bacteria are capable of forming surface-associated, architecturally complex communities, known as biofilms[1,2]. The Gram-positive bacterium *Bacillus subtilis* forms morphologically complex colony biofilms on solid surface and pellicle biofilms at the air/liquid interface, in specialized biofilm-inducing media[3,4]. Cell differentiation occurs in the *B. subtilis* biofilm both spatially and temporally, and is regulated by integrative signaling pathways and influenced by various environmental factors[4–6]. Cell differentiation generates phenotypically distinct cell types within the biofilm, such as matrix producers, swimmers, competent cells, antibiotic producers, and so on. Each cell type possesses unique features and functionality, yet different cell types complement with each other in that the entire community shows synergy and cooperation[6,7]. In summary, complexity and heterogeneity is a hallmark feature of the *B. subtilis* biofilm.

Iron is an essential nutrient element for growth and a cofactor for various enzymes and proteins involved in key biological processes in the bacteria, in particular cellular metabolism and energy generation[8]. In the natural environments, iron availability is very limited due to extremely low solubility of ferric iron ($Fe^{3+}$) under neutral pH ($10^{-18}$ M)[8]. Bacteria thus developed various strategies to uptake iron from the environment. When intracellular levels of iron are low, bacteria turn on multiple iron uptake systems for iron acquisition. In *B. subtilis*, those iron acquisition systems are composed of transporters for the import of elemental iron, ferric citrate, ferrioxamine, ferrichrome, and petrobactin[9]. A key regulator, Ferric uptake regulator (Fur), is responsible for the regulation and derepression of the above iron acquisition systems when iron is limited[10,11]. Fur also regulates genes involved in the biosynthesis of a *B. subtilis* siderophore, bacillibactin (DhbA-CEBF), and a cognate uptake system (FeuABC-YusV)[12,13]. Bacillibactin is a catechol siderophore and binds iron with an extremely high affinity, allowing *B. subtilis* cells to acquire very low amounts of iron from the environment. In most pathogens, siderophores are essential for the bacteria to survive in the host or environment[14–16]. On the other hand, excessive intracellular iron could be toxic due to its involvement in the Fenton's reaction, which generates non-selective free radicals that can damage various biological molecules such as protein, lipid, and DNA[17]. Thus, in bacteria, iron homeostasis plays an important role in maintaining an appropriate range of intracellular iron levels to satisfy the normal need for metabolism and growth[8,15].

Recent studies indicate importance of iron not only in free-living bacteria but also in biofilm formation[18–20]. In biofilm-forming bacteria such as *Pseudomonas aeruginosa* and *Staphylococcus aureus*, high levels of iron were shown to be important for robust biofilm formation[18,19], suggesting a common and important role of iron in biofilm development. In *B. subtilis*, robust biofilm formation also demands unusually high levels of ferric iron, hundreds fold higher than needed for normal growth[3], yet the exact reason is less clear. In this study, we aim to address the above question. We provide mechanistic details of why excessive iron is needed for robust biofilm formation in *B. subtilis* and uncover an adaptive strategy for acquisition and utilization of large amounts of iron during *B. subtilis* biofilm development.

## Results

**2,3-Dihydroxybenzoate is essential for biofilm formation**. In a previous study[21], we investigated the role of various non-ribosomal peptides (NRPs) in *B. subtilis* in biofilm formation. Many of those NRPs had dispensable roles in biofilm formation as assessed by the biofilm phenotypes of the mutants[21]. For instance, two mutants deficient in either bacillaene or fengycin biosynthesis showed little noticeable biofilm phenotypes (Δ*pks*

and Δ*pps*, Fig. 1a, b). Surprisingly, a Δ*dhbA* mutant deficient in the siderophore bacillibactin production was very defective in biofilm formation (Fig. 1b). Bacillibactin is a catechol siderophore that binds to ferric iron with an extremely high affinity[22]. The role of bacillibactin in iron acquisition is well known, but its role in biofilm formation has not been studied in *B. subtilis* until very recently[23].

Bacillibactin biosynthesis relies on the *dhbA-F* operon encoding enzymes that carry out four sequential reactions converting 3-chorismate to bacillibactin (Fig. 1a, c)[24]. Since *dhbF* encodes the most important enzyme involved in the final reaction in bacillibactin biosynthesis (Fig. 1c), a deletion mutant of *dhbF* (Δ*dhbF*) was constructed and the biofilm phenotype of the mutant was examined. To our surprise, the mutant did not exhibit any noticeable biofilm defect (Fig. 1d). Puzzled by this observation, non-polar in-frame deletion mutations for each individual genes in the *dhbA-F* operon were constructed and biofilm phenotypes by those in-frame deletion mutants were examined. As shown in Fig. 1d, Δ*dhbA*, Δ*dhbB*, and Δ*dhbC* all had a very severe biofilm defect. In contrast, Δ*dhbE* formed robust pellicle biofilms, similar to that of Δ*dhbF*. Since DhbE and DhbF are known to be only involved in the last step of bacillibactin biosynthesis, which (together with DhbB) converts 2,3-dihydroxybenzoate (DHB) to bacillibactin, a trimeric ester of 2,3-dihydroxybenzoate-glycine-threonine (Fig. 1c), these results suggest that bacillibactin is not important; rather, the precursor DHB plays an essential role in biofilm formation. To further test this idea, a chemical complementation was performed by supplementing pure DHB (1 µg ml$^{-1}$) to the biofilm media. This time, Δ*dhbA*, Δ*dhbB*, and Δ*dhbC* all formed robust, wild-type-like pellicle biofilms (MSgg + DHB, Fig. 1d). We also noticed that in order to completely rescue the biofilm defect of Δ*dhbA*, Δ*dhbB*, and Δ*dhbC* to the wild-type level, supplementation of about 60 µM (equals to about 1 µg ml$^{-1}$) DHB was needed, implying that wild-type cells produced and secreted DHB sufficiently. This implication was also supported by a recent study, in which secreted DHB amounts were directly measured[23]. DHB is able to bind ferric ion, albeit at a lower binding affinity compared to that of bacillibactin[9]. A similar DHB, 3,4-dihydroxybenzoate, or protocatechuic acid, produced by *Corynebacterium glutamicum*, is a well-studied iron chelator[25,26].

**dhbA-F genes are differentially expressed in biofilm cells**. To learn more about DHB biosynthesis, we decided to investigate the regulation of the *dhbA-F* operon under biofilm conditions by quantitative real-time PCR (RT-qPCR). To do so, a pair of primers were designed to probe transcription for each of the *dhbA*, *dhbB*, *dhbC*, and *dhbE* genes, while three pairs were designed to probe *dhbF* transcription since the size of *dhbF* (~7.13 kb) is almost 60% of the entire operon (Fig. 2a). Results from RT-qPCR showed that in cells collected from pellicle biofilms, expression of all genes in the *dhbA-F* operon was detected except for *dhbF* (black bars, Fig. 2b). In fact, all three pairs of primers failed to detect the transcription of *dhbF* (# indicates below detection limit, Fig. 2b). This result was quite interesting to us for two reasons. First, differential expression of the genes within the *dhbA-F* operon was observed, which could provide genetic basis for strong production of DHB (over bacillibactin) during biofilm formation. In a recent study, Rizzi et al.[23] quantified the production of DHB and bacillibactin during pellicle biofilm formation in *B. subtilis* and found that DHB was produced about 10-fold higher than bacillibactin. Second, the *dhbA-F* operon was readily expressed in the iron-rich biofilm medium (50 µM FeCl$_3$). This seemed to contradict to the knowledge that the *dhbA-F* operon is repressed by Fur under iron-rich conditions in *B.*

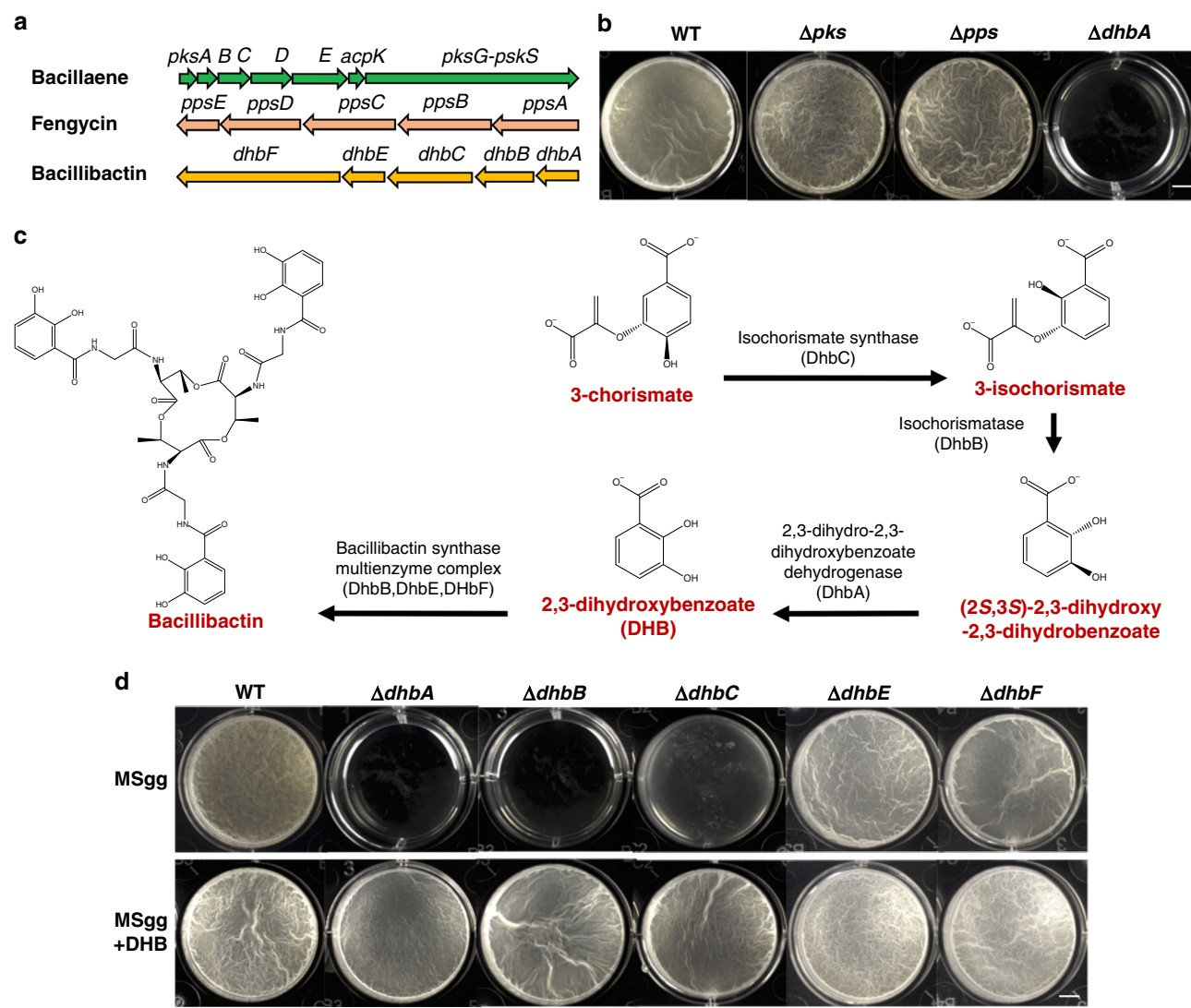

**Fig. 1** 2,3-Dihydroxybenzoate (DHB) is essential for biofilm formation. **a** A diagram of biosynthetic gene clusters for three different non-ribosomal peptides (NRPs) in *B. subtilis*. **b** Formation of pellicle biofilms by the mutants deficient in the biosynthesis of the three NRPs. Strains shown are WT(3610), Δ*pks* (CY167), Δ*pps*(CY168), and Δ*dhbA*(YQ90, insertional deletion). Cells were incubated in MSgg at 30 °C for 2 days before images were taken. Scale bar, 5 mm. **c** A 4-step enzymatic conversion from 3-chorismate to bacillibactin carried out by enzymes encoded in the *dhbA-F* operon[24]. **d** Formation of pellicle biofilms by non-polar in-frame deletion mutants of the *dhbA-F* operon and chemical complementation. Strains shown are WT(3610), Δ*dhbA*(YQ97, in-frame deletion), Δ*dhbB*(YQ98), Δ*dhbC*(YQ99), Δ*dhbE*(YQ100), and Δ*dhbF*(YQ101). Cells were incubated in MSgg at 30 °C for 2 days before images were taken. For chemical complementation, DHB was added at the final concentration of 1 μg ml$^{-1}$ (~60 μM). Scale bar, 5 mm

*subtilis*[10,13]. The qPCR assay was repeated by using pellicle bio-film cells from modified MSgg that contained only 1% of the regularly added ferric iron (0.5 μM FeCl$_3$). Again, it failed to detect any expression of the *dhbF* gene (gray bars, Fig. 2b). Expression of all other genes in the operon was detected, but at a much lower level when compared to that of 50 μM FeCl$_3$. The undetectable expression of *dhbF* under both iron conditions (50 and 0.5 μM of FeCl$_3$) seemed to match with the observed wild-type-like biofilm phenotypes of Δ*dhbF*, suggesting a largely dis-pensable role of bacillibactin in biofilm formation. Lastly, as a note, the expression of *dhbF* was detected when cells were grown under extreme iron-limiting conditions (MSgg without any added FeCl$_3$), and the Δ*dhbF* mutant grew much slower in minimal media without FeCl$_3$ supplementation (data not shown), con-sistent with previous studies[22].

**dhbA-F is regulated by AbrB in response to biofilm signals.** The observation that the expression of the *dhbA-F* operon was much

lower at the lower iron condition (0.5 μM) than the higher iron condition (50 μM) was somewhat surprising given that the *dhbA-F* operon is known to be repressed by Fur under iron-rich con-ditions in *B. subtilis*[13]. Therefore, increased expression of the *dhb* genes under the higher iron condition (50 μM FeCl$_3$) was unlikely due to the regulation by Fur. To search for additional regulators for the *dhbF* operon under biofilm conditions, the regulatory sequence of the operon was analyzed and a putative AbrB recognition motif was identified[27]. AbrB is a biofilm repressor and a transition state regulator known to control a number of operons involved in biofilm formation and secondary metabolites biosynthesis[28–30]. To test if AbrB regulates the operon, a pro-moter-*lacZ* fusion for the *dhb* operon (P$_{dhbA}$-*lacZ*) was con-structed and integrated to the chromosomal *amyE* locus of the wild-type strain, the Δ*abrB* mutant, and the Δ*fur* mutant (as a positive control). Expression of the reporter fusion in the wild-type and the mutants was assayed. The results confirmed that in the Δ*fur* mutant, the activity of the reporter fusion was very

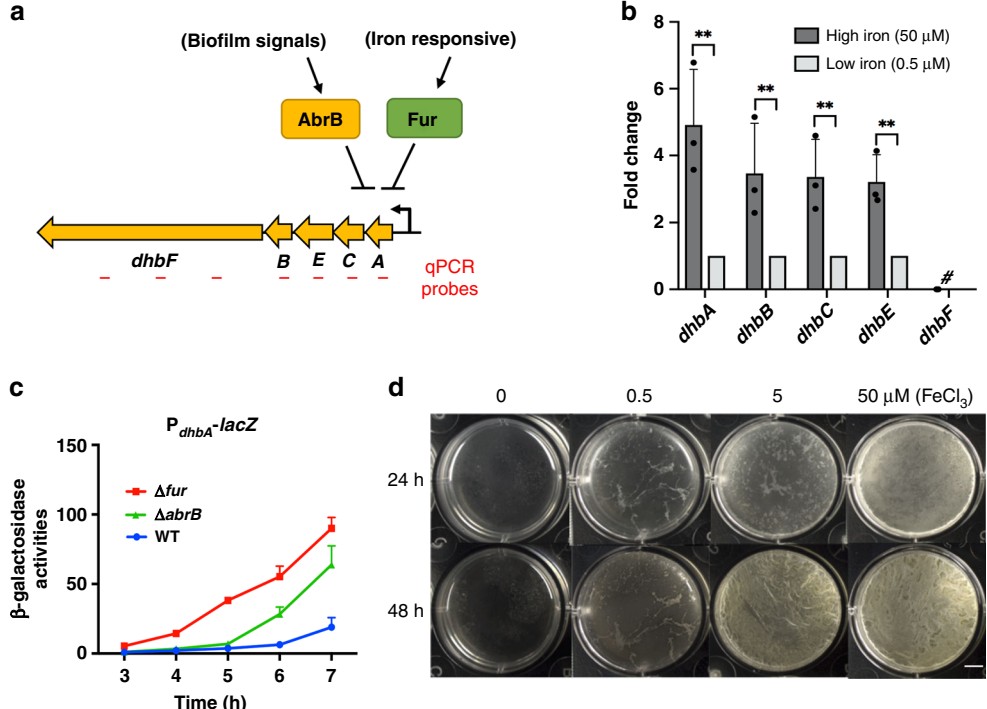

**Fig. 2 a–c** The *dhbA-F* operon is differentially expressed and regulated by AbrB. **a** A diagram demonstrating a putative co-repression of the *dhbA-F* operon by both Ferric uptake regulator (Fur) and the biofilm regulator AbrB. Repression by Fur is known to be iron-responsive, while AbrB-mediated repression is in response to biofilm signals[10,30]. Short lines below represent regions in each of the genes in the *dhbA-F* operon covered by pairs of primers in real-time PCR analyses. Three pairs of primers were designed to probe the *dhbF* gene expression due to its unusually large size. **b** Real-time PCR analyses to probe expression of the genes in the *dhbA-F* operon. Total RNAs were prepared from cells in the pellicle biofilm developed under two different ferric iron supplementations (50 vs. 0.5 μM FeCl₃). Expression of *dhbA*, *dhbB*, *dhbC*, and *dhbD* was found much higher in cells grown in media with 50 μM FeCl₃ than in media with 0.5 μM FeCl₃. The relative transcriptional level in the latter media was set at 1; fold changes represent the fold differences in transcription under two different media conditions. "#" indicates that the transcription of *dhbF* was below detection limit using each of the three pairs of the probes under both media conditions. Data presented are the mean ± s.d. (*n* = 3). Error bars represent standard deviations. Statistical significance was assayed using unpaired *t* test via Prism 6. Stars indicate *P* values <0.01. **c** β-Galactosidase activities of cells bearing the P*dhbA*-*lacZ* transcriptional fusion. Cells were grown in MSgg supplemented with 50 μM FeCl₃ at 37 °C with shaking. Cultures were collected periodically and assayed for β-galactosidase activities. Reporter strains used here are YQ141(WT), YQ142(Δ*fur*), and YQ255(Δ*abrB*). Error bars indicate standard deviations. **d** *Bacillus subtilis* biofilm formation demands unusually high amounts of ferric iron. Pellicle biofilm development by *B. subtilis* 3610 in MSgg supplemented with different amounts of FeCl₃ as indicated. Samples were incubated statically at 30 °C for either 24 or 48 h before images were taken. Scale bar, 5 mm. Source data are provided as a Source data file

strongly induced (Fig. 2c), suggesting that in the wild-type strain, Fur-mediated strong repression was in place. Interestingly, in the Δ*abrB* mutant, the activity of the reporter fusion was also strongly induced (Fig. 2c). This indicates that under biofilm conditions, the *dhb* operon is co-repressed by both AbrB and Fur and derepression can be triggered by either Fur in response to iron limitation or by inactivation of AbrB via sensing of biofilm signals (Fig. 2a). Previous studies have shown that AbrB was gradually inactivated in the presence of biofilm signals through the known biofilm pathway and the master regulator Spo0A[30,31]. The observed modest derepression of the *dhbA-F* operon in wild-type cells in later time points (blue line, Fig. 2c) was consistent with the above idea and with the expression profile of other AbrB-regulated genes such as the biofilm matrix operon *tapA-sipW-tasA*[30].

**Biofilm formation demands large amounts of ferric iron.** Although ferric iron is very poorly soluble in regular media, large amounts of ferric iron were shown to be needed in the biofilm media for *B. subtilis* to form robust biofilms (Fig. 2d). A decrease of the concentration of ferric iron in the media from 50 to 5 μM already started to impact biofilm formation while a further decrease to 0.5 μM severely blocked biofilm formation (Fig. 2d).

The block in biofilm formation was not due to growth inhibition, since only a mild difference was seen in growth even if ferric iron was not at all added to the media (presumably trace amount of ferric iron was present in the water and other chemical ingredients during media preparation) (Supplementary Fig. 1A).

**Excess iron stimulates production of iron-binding enzymes.** Higher media iron concentrations might boost yet unknown activities critical for robust biofilm formation. Many of the iron-responsive genes are under the control of Fur[9,32]. We argued that those genes would be less likely to be directly involved since many of them are primarily involved in iron uptake and since those genes would be further repressed by Fur upon increasing iron concentrations[10]. Our focus was then shifted to genes encoding proteins that contain iron as a cofactor. Activities of those proteins depend on iron binding and thus cellular iron availability[8]. One category of those proteins consists of enzymes involved in cellular metabolism and energy generation, such as glycolysis, citric acid cycle, and electron transfer chain (ETC) (Fig. 3a, b). Previous studies have shown that the abundance of some of those enzymes increased significantly in response to elevated intracellular iron availability[33]. It was unclear whether increased abundance of those proteins was due to upregulated gene expression

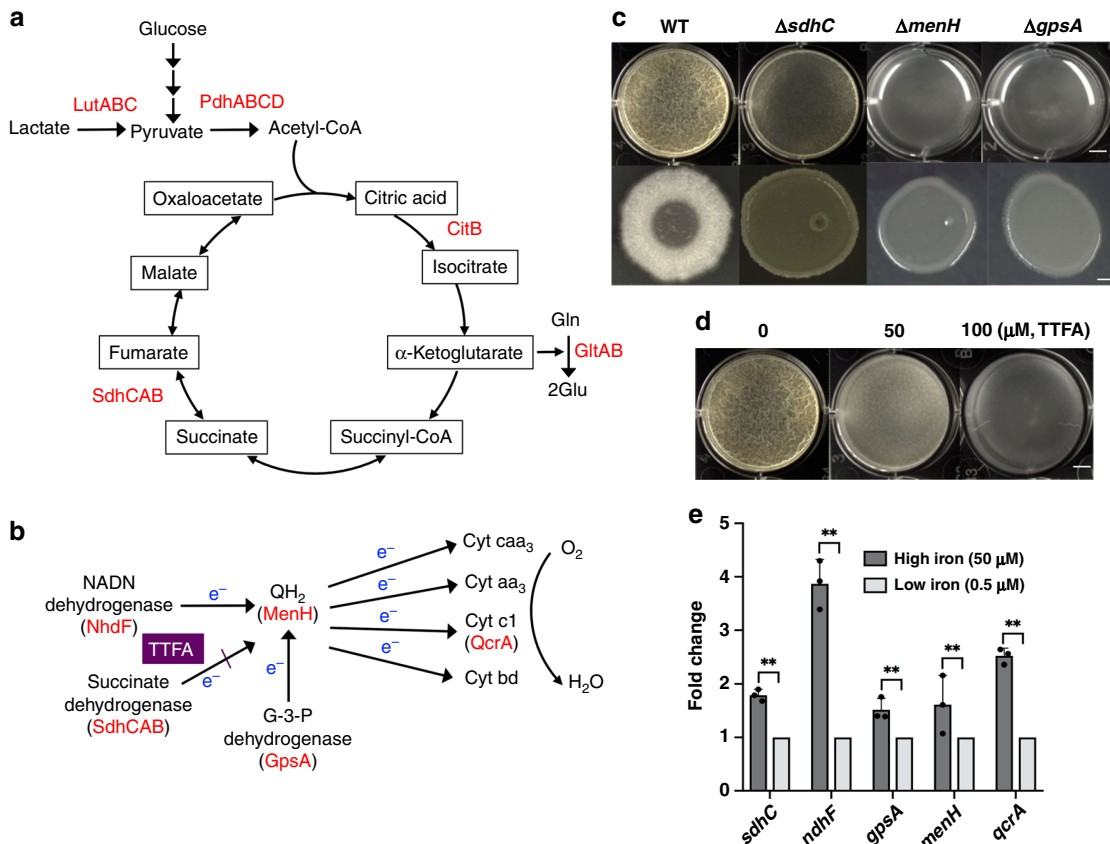

**Fig. 3** High iron concentrations stimulate production of iron-binding metabolic enzymes. **a** The diagram of glycolysis and citric acid cycle in *B. subtilis*. Highlighted in red are enzymes whose activities depend on iron as a cofactor. LutABC, lactate utilization protein complex[34]; PdhABCD, pyruvate dehydrogenase complex[67]; CitB, citric acid isomerase[67]; SdhCAB, succinate dehydrogenase[67]. **b** The diagram of the electron transfer chain in *B. subtilis*. Highlighted in red are enzymes whose activities depend on iron as a cofactor. NADH dehydrogenase, glycerol-3-phosphage (G-3-P) dehydrogenase, and succinate dehydrogenase are three primary substrate dehydrogenases denoting electrons to the electron transfer chain[67]. The electron transfer from succinate dehydrogenase to the electron carrier quinone can be blocked by a chemical inhibitor thenoyltrifluoroacetone (TTFA)[37]. **c** Pellicle and colony biofilm formation by the wild-type and three metabolic mutants of *B. subtilis*, Δ*sdhC*, Δ*menH*, and Δ*gpsA*. Cells were incubated in MSgg at 30 °C for 2 days before images were taken. Scale bar at the top panel, 5 mm; Scale bar at the bottom panel, 2.5 mm. **d** Inhibition of *B. subtilis* pellicle biofilm formation by TTFA. Cells were incubated in MSgg at 30 °C for 2 days before images were taken. TTFA was added at the final concentration of 0, 50, or 100 μM at the beginning of inoculation. Scale bar, 5 mm. **e** Real-time PCR analyses to probe expression of the genes involved in electron transfer. Total RNAs were prepared from cells collected from the pellicle biofilm in MSgg but with two different ferric iron supplementations (50 vs. 0.5 μM FeCl₃). Expression of *sdhC*, *ndhF*, *gpsA*, *menH*, and *qcrA* was found much higher in cells in media with 50 μM FeCl₃ than in media with 0.5 μM FeCl₃. The relative transcriptional level in the latter media was set at 1; fold changes represent the fold differences in transcription under two different media conditions. Data presented are the mean ± s.d. (*n* = 3). Error bars represent standard deviations. Stars indicate *P* values <0.01. Statistical significance was assayed using unpaired *t* test via Prism 6. Stars indicate *P* values <0.01. Source data are provided as a Source data file

upon increasing iron availability, and more importantly whether any of those proteins might be critical for biofilm formation. To test that, three different experiments were carried out. In the first experiment, a number of genes encoding metabolic enzymes containing iron as a cofactor were selected (highlighted in red, Fig. 3a, b). Deletion mutants for each of those genes were constructed and biofilm phenotypes of the mutants were examined. Some mutants had no noticeable biofilm phenotype, such as Δ*lutABC*[34] and Δ*citB* (Supplementary Fig. 2A), while some others, although successfully constructed, were severely impaired in growth, such as Δ*gltA*, Δ*nhdF*, and Δ*qcrA* (data not shown). Those mutants were not further investigated. The remaining three mutants, Δ*sdhC*, Δ*menH*, and Δ*gpsA*, were most interesting to us since they all showed a very severe biofilm defect (Fig. 3c), and no noticeable growth defect compared to the wild type (Supplementary Fig. 1B). Interestingly, all three enzymes are known to be involved in respiratory electron transfer (Fig. 3b), while SdhC, part of the succinate dehydrogenase complex, is also

involved in citric acid cycle (Fig. 3a)[35]. These results indicated that although impaired electron transfer had much less of an impact on growth under tested conditions, it was detrimental to biofilm formation in *B. subtilis*.

In the second experiment, expression of selected genes (*sdhC*, *menH*, *gpsA*, *nhdF*, and *qcrA*) encoding for enzymes involved in electron transfer[36] was investigated by RT-qPCR. Cells were collected from pellicle biofilms developed under two different media ferric iron conditions (0.5 vs. 50 μM FeCl₃). An elevated induction in expression of all five genes was observed under the 50 μM media ferric iron concentration, compared to that of 0.5 μM (Fig. 3e). When the qPCR assay was performed using cells collected under shaking conditions, an even stronger induction for all five genes was observed under the 50 μM media ferric iron concentration, compared to that of 0.5 μM (ranged from 10.5- to 28.3-fold increase, Supplementary Fig. 2B). These results suggested that the abundance of those proteins could increase significantly upon increasing concentrations of iron in the growth

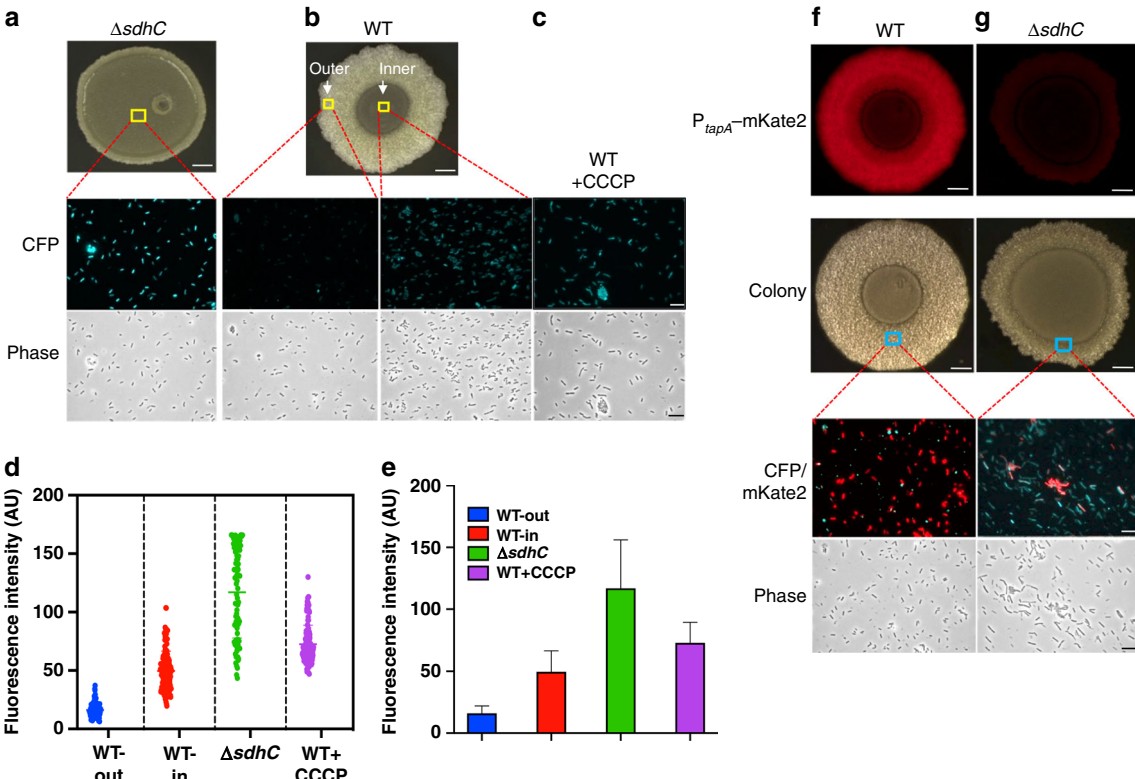

**Fig. 4** Membrane potential is key to biofilm matrix production. **a** Detection of the membrane potential in cells collected from the inner region of (indicated by the yellow square) a 2-day Δ*sdhC* biofilm colony by using the fluorescent dye thioflavin T (ThT, shown as cyan fluorescent protein (CFP)). The overall strong cyan fluorescence (CFP) signal of the Δ*sdhC* cells implies weak membrane potential of the cells since the dye accumulation anticorrelates with the membrane potential. **b** Detection of the membrane potential in cells collected from the inner or the outer region (indicated by yellow squares) of a 2-day wild-type biofilm colony by using the fluorescent dye ThT. **c** Detection of the membrane potential in wild-type cells from shaking culture treated with the membrane depolymerizing agent CCCP (1 mM) as a control[41]. **d, e** Distribution (**d**) and quantification (**e**) of average pixel density from the ThT fluorescence dye accumulation in the cells from the outer and inner regions of the wild-type colony biofilm, Δ*sdhC* colony biofilm, and from wild-type cells treated with CCCP. Data presented are the mean ± s.d. (*n* = cell number in each image). Error bars represent standard deviations. **f, g** Simultaneous detection of the matrix gene expression and membrane potential in cells collected from the wild-type (**f**) or the Δ*sdhC* mutant (**g**) colony biofilms. The fluorescence reporter P*tapA*-mKate2 was used to probe the expression of the matrix operon *tapA-sipW-tasA*. Membrane potential was probed by using the ThT dye (shown as CFP). Biofilm colonies with red fluorescence were recorded by using a dissecting fluorescent microscopy. Scale bars in all images showing biofilms, 5 mm. Scale bars in all microscopic images showing individual cells, 10 μm. Source data are provided as a Source data file

media. In the third experiment, we tested if blocking electron transfer by using small inhibitory chemicals could similarly impact biofilm formation in *B. subtilis*. 2-Thenoyltrifluoroacetone (TTFA), a small chemical inhibitor known to block the electron transfer from succinate dehydrogenase to the electron carrier quinone (Fig. 3b), was applied[37]. As shown in Fig. 3d, TTFA inhibited robust biofilm formation at concentrations (e.g., 50 μM) that did not significantly impact the growth of the cells (Supplementary Fig. 1C). In summary, these results suggest that media iron concentrations have a strong effect on the expression of iron-containing ETC proteins, and that some of those proteins are essential for robust biofilm formation in *B. subtilis*.

**Membrane potential is key to biofilm matrix production**. One of the key activities of the ETC is to establish strong membrane potential. Because of the role of succinate dehydrogenase (complex II) in the ETC[38], we speculated that the Δ*sdhC* mutant may have impaired membrane potential. To test that, cells were collected from the biofilm colony of Δ*sdhC* and membrane potential of the cells was measured by using the dye thioflavin T (ThT, Fig. 4a). According to previously published studies[39,40], accumulation of this fluorescent dye inside the cells and thus the fluorescent density of the cells anti-correlate with the membrane potential of the cells. As shown in Fig. 4a, the majority of cells

collected from the Δ*sdhC* colony biofilm demonstrated strong fluorescence (shown as cyan fluorescent protein (CFP)), indicating a low membrane potential of the cells. Cells were also collected from the wild-type colony biofilm (the outer region, Fig. 4b). In contrast, the majority of those wild-type cells demonstrated very weak fluorescence, indicating high membrane potentials of the cells (Fig. 4b). Interestingly, when cells were collected from the inner region of the wild-type colony biofilm, those cells showed intermediate fluorescence, suggesting that they had relatively lower membrane potentials compared to the cells in the outer region of the same colony biofilm (Fig. 4b). Quantitative analyses of the fluorescence signals of individual cells were performed in parallel. The results further demonstrated the significant difference in membrane potential between the wild-type and the Δ*sdhC* cells, and even between cells in the outer and inner regions of the wild-type colony biofilm (Fig. 4d, e). Lastly, a known membrane depolarizing agent, carbonyl cyanide *m*-chlorophenylhydrazine (CCCP)[41], was applied to validate the ThT fluorescent dye-based analyses of membrane potential. The result showed that the addition of CCCP (1 mM) significantly impaired membrane potential of the wild-type cells as indicated by the strong fluorescence inside the cells (Fig. 4c), but slightly less severe than the impairment caused by Δ*sdhC* (Fig. 4d, e).

Previous studies have shown that cells in the outer region of a colony biofilm tended to be robust matrix producers, while cells in the inner region were often old and weaker in matrix production[42,43]. Thus, a positive correlation between membrane potential and matrix production of the cells seemed to exist. To test if there is such a potential correlation, a dual-labeling technique was applied in that the ThT dye (shown as CFP) was used as an indicator for the membrane potential of the cells while cells also contained a $P_{tapA}$-$mKate2$ fluorescent fusion to report the expression of the key biofilm matrix operon $tapA$-$sipW$-$tasA$[30]. Results from the assay showed that in the wild-type colony biofilm, cells from the outer region had both strong expression of the matrix gene reporter ($mKate2$) and strong membrane potential (which anticorrelates with CFP signals) (Fig. 4f). Cells from the $\Delta sdhC$ colony biofilm showed the opposite in that the majority of those cells were both weak in matrix gene expression and in membrane potential (Fig. 4g). Our results thus far supported the idea that membrane potential is closely linked to matrix gene expression and biofilm robustness in B. subtilis.

**Large amounts of soluble iron are matrix associated**. In all the experiments described above, when concerning iron concentrations, they were always referred to the amounts of ferric iron in the media. This was not satisfactory first because ferric iron was largely insoluble in the media, and second the intracellular concentrations of iron was unknown. Thus, we decided to investigate the intracellular concentrations of iron when cells were grown under two different media ferric iron concentrations (either 50 or 0.5 μM $FeCl_3$) by using inductively coupled plasma-mass spectrometry (ICP-MS). Surprisingly, results obtained from the assay showed that although iron concentrations in the media differed by 100-fold, the intracellular total iron concentrations differed by just less than one-fold (~70%, Fig. 5a). It was more surprising to recall that when 0.5 μM $FeCl_3$ was provided in the media, B. subtilis biofilm was severely impaired (Fig. 2d).

We had hypothesized on (i) either extracellular association of large amounts of soluble irons in the biofilm or (ii) heterogeneous distributions of iron within different regions of the cells in the biofilm. To test the above hypotheses, cells from either the inner or the outer region of the wild-type colony biofilm were picked. Both the intracellular iron concentration and the concentration of soluble iron associated extracellularly with the biofilm were measured similarly using ICP-MS (Fig. 5b). For the latter, a protocol was previously published by us to allow separation of cells from the extracellular matrix in a biofilm[44]. The most striking result obtained from those iron measurements was that large amounts of soluble irons (precipitated iron was removed by filtration) were found associated with the biofilm matrix for the cells collected from the outer region of the colony biofilm, more than 10-fold higher than the intracellular concentration of iron from the same cells (Fig. 5b). In contrast, the difference in the intracellular iron concentrations in the cells from the outer or center region was rather modest (~50%, Fig. 5b). These results depicted two different scenarios for possible roles of ferric iron in promoting biofilm formation, one being internalized and boosting the activities of iron-containing enzymes involved in electron transfer for establishment of strong membrane potential (as shown in Figs. 3e, 4a–e), and the other being solubilized (potentially by secreted DHB) and extracellularly associated with the biofilm matrix (as shown in Fig. 5b).

**Extracellular iron could be involved in electron transfer**. Although it is possible that those extracellularly associated ferric iron function to facilitate matrix assembly, we speculated that those ferric irons were involved in respiratory electron transfer but extracellularly, either as terminal receptors or shuttles among different cells[45]. The rationale for the second possibility is that efficient electron transfer normally depends on molecular oxygen as the terminal receptor; however in the biofilm, internal layers of cells likely lack the access to molecular oxygen (which we show next). For those cells to perform electron transfer and establish membrane potential, they will need help of alternative electron receptors intracellularly and/or extracellularly.

To test this possibility, three experiments were performed. In the first experiment, accessibility to molecular oxygen by cells in different depth of a colony biofilm was tested by using an automated oxygen microelectrode piercing through the colony biofilm (see Methods). Both the outer and the center regions of the colony biofilm were chosen for oxygen measurement (Fig. 5b). In the outer region, there was a decline in oxygen levels from the top to the bottom of the colony biofilm (blue line, Fig. 5c) and the rate of oxygen depletion was initially slow but accelerated when reaching about 1/3 of the biofilm depth (−120 μm from the surface). In the center, the pace of oxygen depletion was more or less linear (red line, Fig. 5c). This result in general supported the idea that cells in the bottom of the colony biofilm may have difficulty in access to oxygen and performing oxidative respiration.

In the second experiment, the redox potential distributed along the vertical axis of the biofilm was measured. For technical reasons, this time we chose the pellicle biofilm formed in a beaker in order to increase the depth of the biofilm to suit the redox microelectrode used in the assay (Fig. 5d). Decreasing redox potential from the top to bottom along the vertical axis of a pellicle biofilm was observed (stronger reduction in the bottom, Fig. 5d). This result correlated with decreasing access of the cells to molecular oxygen (Fig. 5c). However, caution may be needed here since measurement of redox potential and that of molecular oxygen were performed in two different biofilm settings (pellicle and colony biofilms, respectively). Changing of redox potential implied that extracellular redox reaction (electron transfer) might happen in the bottom layers of the pellicle biofilm. We further probed the membrane potential status in parallel in the cells at the top and bottom layers of the pellicle biofilm. A published protocol[46] by us was modified to allow separation of cells at the top and bottom layers of a pellicle biofilm by applying a metal mesh in between (see Methods) (Fig. 5e). Cells at the top and bottom were separately collected and measured for membrane potential using the ThT fluorescence dye as described above. Cells at the top demonstrated strong membrane potential as indicated by very weak fluorescence (Fig. 5f). Interestingly, cells at the bottom also showed relatively strong membrane potential (Fig. 5f), implying active respiratory electron transfer in those cells even without sufficient access to molecular oxygen. We speculated that respiratory electron transfer and establishment of membrane potential in those cells relied on large amounts of matrix-associated extracellular irons as the electron receptors or shuttles. Lastly, we also looked at the viability of the bottom (and the top) layers of cells in the pellicle biofilm by using live/dead staining. The result showed that cells at the bottom of the pellicle biofilm had a similar live/dead ratio when compared to the cells at the top (Supplementary Fig. 3).

In the third experiment, in order to test if extracellular electron transfer (EET) does occur in the B. subtilis biofilm, a cyclic voltammetry (CV) measurement (Fig. 6a) was performed by coating the wild-type (WT) biofilm on a working reticulated vitreous carbon (RVC) foam electrode. As shown in Fig. 6b, two redox peaks appeared with potentials at around −40/−170 mV (pointed by arrows in the red curved lines). This indicates that the biofilm coated on the RVC foam electrode can reduce $Fe^{3+}$ to

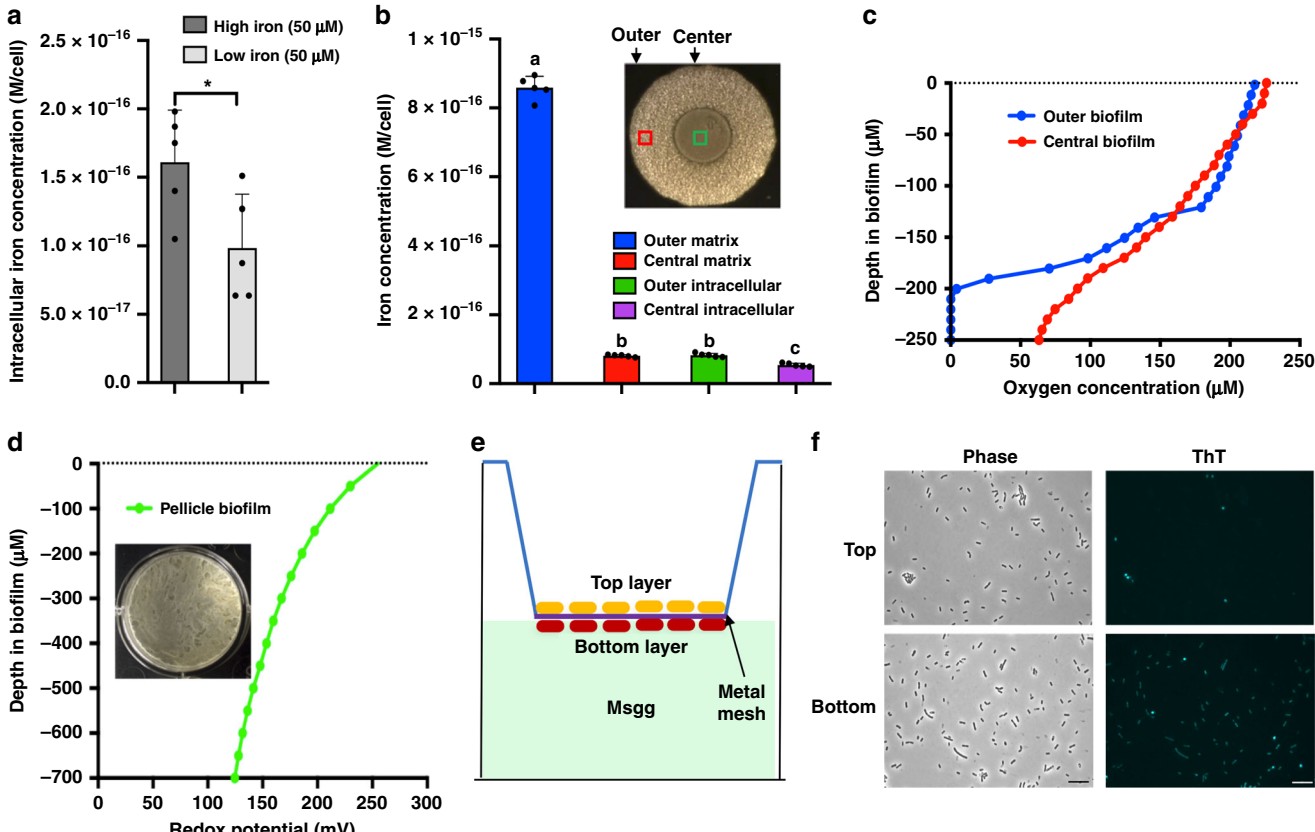

**Fig. 5** Large amounts of matrix-associated ferric iron could act as extracellular electron receptors. **a** Measurement of intracellular iron concentration by inductively coupled plasma-mass spectrometry (ICP-MS). Samples were prepared from cells grown at 37 °C by shaking under two different media conditions: MSgg supplemented with either 50 or 0.5 μM of FeCl₃. Intracellular total iron concentration was indicated as molar per cell. Data presented are the mean ± s.d. ($n = 5$). Error bars represent standard deviations. Statistical significance was assayed using unpaired $t$ test via Prism 6. Star indicates $P$ value <0.05. **b** Measurement of intracellular and extracellular (matrix-associated) iron concentrations for cells collected from either the inner or the outer region (indicated by the red and green squares) of a wild-type biofilm colony. The biofilm colony was developed in regular MSgg agar media (50 μM FeCl₃) at 30 °C for 2 days prior to collecting cells. Data presented are the mean ± s.d. ($n = 5$). Error bars represent standard deviations. Statistical significance was assayed using unpaired $t$ test via Prism 6. The same letter on the bars of each column indicates no significant difference. **c** Measurement of molecular oxygen concentrations along the vertical depth of the colony biofilm. Both the inner (red line) and the outer (blue line) region of the colony biofilm were picked for analysis. **d** Measurement of redox potential along the vertical depth of the pellicle biofilm. **e** A cartoon demonstration of the setting to use a metal mesh (Nalgene) to separate top and bottom cells in a pellicle biofilm. **f** Cells from the top and bottom of a pellicle biofilm were collected and assayed for membrane potential status using the fluorescence dye ThT as described previously. Scale bar, 10 μm. Source data are provided as a Source data file

Fe²⁺ in the aqueous solution. In the two control measurements, one with the biofilm medium MSgg only and the other with MSgg supplemented with 1 μg ml⁻¹ of DHB, no clear redox peak was observed (Fig. 6b).

## Discussion

Iron is an essential nutrient element for growth in almost all organisms. A unique question investigated in this study is why excessive amounts of ferric iron, hundreds fold higher than for normal growth, are needed to promote robust biofilm formation in the bacterium *B. subtilis*. Two important roles of iron in promoting *B. subtilis* biofilm formation are proposed in this study. For one, large amounts of iron were found associated extracellularly with the biofilm matrix. Results from electrochemical assays suggested that EET occurred in the *B. subtilis* biofilm and those matrix-associated extracellular irons could be involved in the EET by potentially acting as electron receptors and shuttles. EET may be especially important for cells located in the deep layers of the biofilm and thus blocked from access to molecular oxygen. For the other role of iron, our evidence supported the idea that a small proportion of iron was internalized and played a critical role in boosting the production of multiple

iron-containing enzymes essential for intracellular electron transfer (IET) during oxidative respiration. Both the EET and IET could help to establish strong membrane potential, which we then showed, was linked to strong matrix gene expression during *B. subtilis* biofilm formation.

Biofilms are considered a very heterogenous environment[47]. Cells in different spatial locations in the biofilm may adopt different strategies for respiration and energy generation. Here we propose a model on respiratory electron transfer during *B. subtilis* biofilm development (Fig. 7). For cells in the upper layers where oxygen is sufficient, they perform respiratory IET using molecular oxygen as the terminal electron acceptor and electrons being donated to the ETC[48] by various substrate dehydrogenases. For cells in the bottom layers of the biofilm where oxygen becomes very limited or even absent, they perform EET using extracellular matrix-associated iron as the terminal electron acceptor. In light of the above model, we would like to point out several limitations in this study. First, evidence on the presence of large amounts of extracellular iron in the deep layers of the biofilm (instead of the entire biofilm) is still needed to support the above model. Second, for redox potential measurement, only pellicle biofilm (but not colony biofilm) was applied since the measurement of redox

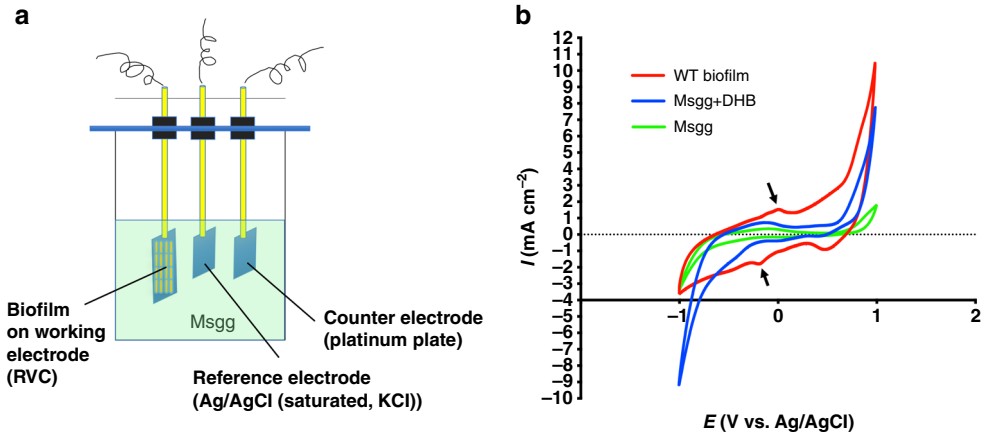

**Fig. 6** Cyclic voltammetry (CV) analysis indicates electron transfer in the *B. subtilis* biofilm. **a** A diagram of the biofilm CV analysis by using an electrochemical workstation equipped with three-electrode system. A reticulated vitreous carbon (RVC) foam working electrode was coated with the pellicle biofilm on its surface, while a platinum plate (1 cm × 1 cm) and Ag/AgCl (saturated, KCl) were used as the counter and reference electrode, respectively. **b** Profile of CV curves. Two redox peaks appeared with potential at around −40/−170 mV (indicated by arrows on the red curved line), indicating that the biofilm coated on RVC foam electrode can reduce $Fe^{3+}$ to $Fe^{2+}$ in aqueous solution (MSgg medium), while the two control assays, one with MSgg medium only and the other MSgg plus DHB, did not show any clear redox peak (blue and green curved lines). Source data are provided as a Source data file

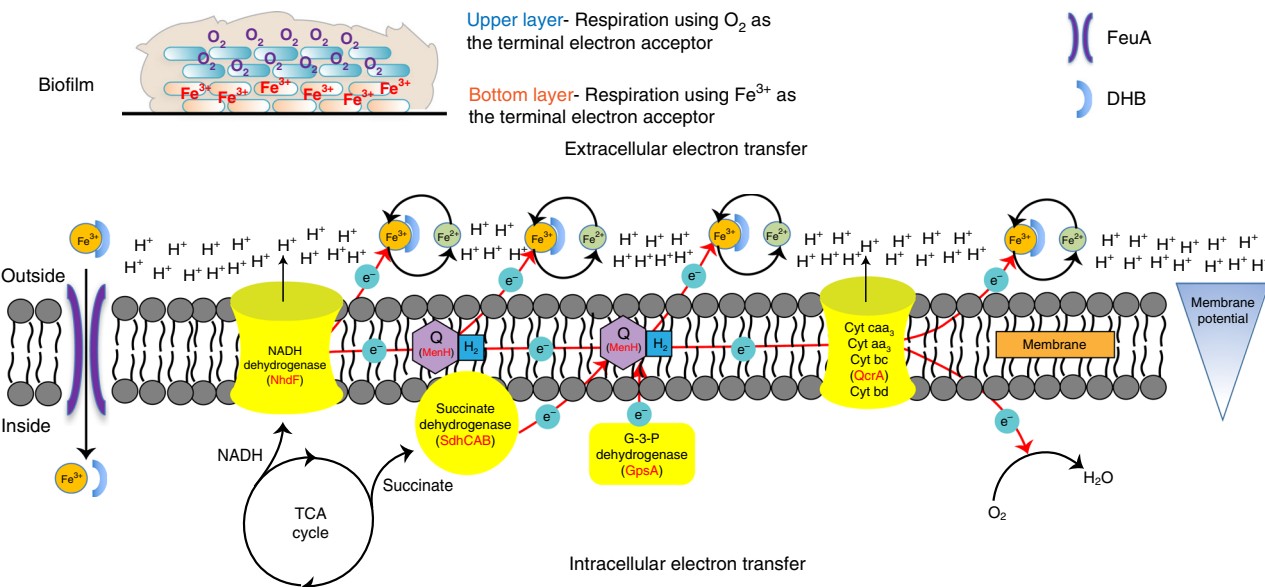

**Fig. 7** A working model for extracellular and intracellular electron transfer in the *B. subtilis* biofilm. Based on this model, *B. subtilis* cells in the upper and bottom layers of the biofilm use different electron acceptors. For cells in the upper layers where oxygen is sufficient, cells use molecular oxygen as the terminal electron acceptor to perform direct intracellular electron transfer (IET), while for cells in bottom layers where oxygen becomes very limited, cells perform extracellular electron transfer (EET) using extracellular matrix-associated iron as the terminal electron acceptor. In the biofilm environment, *B. subtilis* is an electrochemically active microorganism (EAM) and likely takes advantage of both IET and EET to perform respiration and establish membrane potential. During EET, ferric iron could get free electron from any of the components in the electron transfer chain, including various initial substrate dehydrogenases, quinone, or cytochromes. The precursor 2,3-dihydroxybenzoate (DHB), but not bacillibactin, is an iron-binding molecule for iron solubilization and acquisition under iron-rich conditions. DHB–iron complex will be taken into the cells via the FeuABC-YusV-mediated transport system[9]

needs aqueous environment to form a circuit between the redox probe and the reference probe and only pellicle biofilm can satisfy this requirement. As colony and pellicle biofilms are two different settings, caution needs to be taken when accessing the results obtained from either colony or pellicle biofilms alone.

In the biofilm environment, *B. subtilis* can be considered an electrochemically active microorganism (EAM and likely takes advantage of both IET and EET to perform respiration in order to establish membrane potential and generate energy. EAMs are microorganisms that can transfer electrons from cells to extracellular electron acceptors such as minerals, contaminants, and electrode[45,49,50]. So far, more than 100 EAMs have been isolated or identified[49]. Most of them are Gram-negative bacteria and belong to *Proteobacteria* phylum[51]. Most studies on the EET mechanism are based on *Shewanella* spp. and *Geobacter* spp.[52,53]. More recently, Gram-positive bacteria *B. megaterium* and *Enterococcus faecalis* also showed the capacity of EET[54,55]. Studies have shown that oxygen accessibility is clearly a key factor driving biofilm heterogeneity. Here we present another interesting example of metabolic heterogeneity in cells within the *B. subtilis* biofilm.

In this study, a link between membrane potential and strong matrix gene expression was demonstrated during *B. subtilis* biofilm formation. Traditionally, the importance of membrane potential has been discussed in the context of ATP generation, bacterial motility, chemotaxis, molecule transport, and cell division. In recent studies, membrane potential has been demonstrated to be important for the electrical communication within cells during the development of bacterial communities in *B. subtilis*[39,56]. However, why membrane potential is essential for biofilm formation in *B. subtilis* is still unclear. Previous studies have suggested the role of membrane potential in transport of key metabolites such as glutamate[40]. Glutamate is a charged amino acid, whose uptake has been shown to depend on ion gradient and membrane potential[57]. Glutamate is a key component in the biofilm-inducing media for *B. subtilis*[3] and its absence will result in structurally much weaker biofilms[58].

We also discovered an adaptive strategy for iron acquisition during *B. subtilis* biofilm formation. Our evidence suggested that in order to utilize necessary amounts of ferric iron in the media, *B. subtillis* cells preferentially produced and relied on the siderophore precursor DHB (a monomeric intermediate of the trimeric bacillibactin) instead of the final siderophore bacillibactin. We argued that this could be an important adaptive strategy for the bacterium for several reasons. First, since the size of the last *dhbF* gene is very large (7.13 kb), expressing this gene and translating the protein may be very costly and possibly error prone. Second, although bacillibactin has an extremely high binding affinity toward ferric iron, uptake and then release of free iron into the cytosol of the cell is not straightforward. The release of free iron needs an extra hydrolysis step due to extremely high binding affinity of bacillibactin and the hydrolyzed bacillibactin is not reusable[59]. Therefore, alternating production of DHB and bacillibactin depending on iron availability in the environment seems to be an important adaptive strategy. In fact, in other species such as *E. coli*, the homologous gene of *dhbF*, *entF*, is separated from other *ent* genes, while clustered with the *fes* gene encoding an enterobactin hydrolase (Supplementary Fig. 4)[60]. This provides a possible genetic basis for alternative production of the two iron-binding molecules in other bacteria[60]. The putative mechanism for differential expression of the *dhbA-F* operon is unknown. We suspect that a transcription attenuation-like regulation between *dhbF* and the gene upstream of *dhbF* may be responsible for substantially attenuated expression of *dhbF* under iron-rich conditions.

## Methods

**Strains and media.** Strains and plasmids used in this study are listed in Supplementary Table 1. *Bacillus subtilis* strain PY79, 168, NCIB 3610[3,27,61], and derived strains were cultured in lysogenic broth[62] at 37 °C. Pellicle biofilm formation in *B. subtilis* was induced using MSgg broth (5 mM potassium phosphate and 100 mM Mops (3-(*N*-morpholino)propanesulfonic acid) at pH 7.0 supplemented with 2 mM MgCl₂, 700 μM CaCl₂, 50 μM MnCl₂, 50 μM FeCl₃, 1 μM ZnCl₂, 2 μM thiamine, 0.5% glycerol, and 0.5% glutamate)[3] and colony biofilm formation was induced in *B. subtilis* using MSgg solidified with 1.5% (w/v) agar at 30 °C. Enzymes used in this study were purchased from New England Biolabs. Chemicals and reagents were purchased from Sigma or Fisher Scientific. Oligonucleotides were purchased from Eurofins Genomics and DNA sequencing was also performed at Eurofins Genomics. Antibiotics, if needed, were applied at the following concentrations: 10 μg ml⁻¹ of tetracycline, 1 μg ml⁻¹ of erythromycin, 100 μg ml⁻¹ of spectinomycin, 20 μg ml⁻¹ of kanamycin, and 5 μg ml⁻¹ of chloramphenicol for transformation in *B. subtilis* and 100 μg ml⁻¹ of ampicillin for *E. coli* DH5α transformations.

**Strain construction.** All the insertional deletion mutants used in this study (listed in Supplementary Table 1) in the *B. subtilis* 168 background were purchased from the Bacillus Genetic Stock Center (http://www.bgsc.org) and introduced into NCIB 3610 via transformation following a previously described protocol[63]. In order to make marker-less deletion in the bacillibactin biosynthetic genes, the *dhbA*, *dhbB*, *dhbC*, *dhbE*, and *dhbF* insertional deletion strains marked with an *erm*-HI resistance cassette in the *B. subtilis* 3610 background were transformed with the

plasmid pDR244 (a temperature-sensitive suicide plasmid with a constitutively expressed Cre recombinase gene)[64]. Transformants were selected on LB agar plates supplemented with spectinomycin at 30 °C (permissive temperature). Transformants were then streaked onto LB agar plates and plates were incubated at 42 °C (non-permissive temperature). Cells from single colonies were spotted onto LB plates, LB plates supplemented with erythromycin and lincomycin, and LB plates with spectinomycin. A transformant that grew only on the LB agar plate but not on LB agar plates supplemented with any of the three antibiotics, likely both lost the *erm* antibiotic resistance cassette and was cured for the pDG244 suicide plasmid[64]. Those transformants were selected and verified for in-frame deletion using PCR. To construct reporter strains with the P*dhbA*-*lacZ* fusion, the promoter sequence of the *dhbA-F* operon was amplified by PCR by using the primers P*dhb*-F and P*dhb*-R (all primers are listed in Supplementary Table 2) and the genomic DNA of 3610. PCR products were then cloned into the vector pDG1728 to make a P*dhbA*-*lacZ* fusion. The integration of the P*dhbA*-*lacZ* fusion or the P*tapA*-*mKate2* reporter fusion, which was from the strain TMN503[65] into the chromosomal *amyE* locus of virous *B. subtilis* strains and verification of such integrations were described in a previous publication[66].

**Microscopic imaging.** For imaging of colony and pellicle biofilms, a Leica MSV269 dissecting microscope with a Leica DFC2900 camera and ×4 magnification was used. Same exposure and acquisition settings were applied to all colony and pellicle biofilm samples. For single-cell fluorescence imaging, cells were cultured as described above. After incubation for indicated times, colony biofilms were collected, resuspended in phosphate-buffered saline (PBS), and disrupted with vortexing. One milliliter of cell resuspension was mildly sonicated with the 5-s pulse at 1.5 output scale for three times (Branson, Model W185). Cells were centrifuged at 14,000 rpm for 1 min and washed briefly with PBS. For imaging, 1 μl of the resuspension was placed on a 1% (w/v) agarose pad and covered with a cover slip. Cells from three independent biological replicates were imaged using a Leica DFC3000 G camera on a Leica AF6000 microscope. Non-specific background fluorescence was determined by quantifying wild-type cells with no reporter fusion. Imaging of samples collected from different time points was conducted using the same exposure settings. For observation of the ThT fluorescence dye, the setting for observing CFPs was used, with the excitation wavelength at 426–450 nm and the emission wavelength at 502–538 nm. For observation of mKate2, the excitation wavelength was set at 540–580 nm and the emission wavelength at 610–680 nm.

**Biofilm assays.** For colony biofilm formation, cells were grown to exponential phase in LB broth and 2 μl of the culture was spotted onto MSgg media solidified with 1.5% (w/v) agar. The plates were incubated at 30 °C for 2 days. For pellicle biofilm formation, cells were grown to exponential phase in LB broth, and 3 μl of the culture was inoculated into 3 ml of MSgg liquid media in a 12-well microtiter plate (VWR). For treatment with DHB, TTFA, or CCCP, the chemical compound was diluted and added to the medium at the indicated final concentrations. Images of colony and pellicle biofilms were taken as described above using a Leica MSV269 dissecting scope and a Leica DMC2900 camera. For separation of the top and bottom cells in pellicle biofilms, pellicle biofilms were developed as described above but with a piece of metal mesh (Nalgene) at the surface of the biofilm liquid media. After 2 days of incubation, the metal mesh was lifted, pellicle biofilms cells attached to the top and bottom of the metal mesh were collected, and cells were then treated as described for microscopic analyses.

**Assays of β-galactosidase activities.** Cells were cultured in MSgg medium at 30 °C with shaking. One milliliter of culture was collected at each indicated time point and cells were centrifuged down at 5000 rpm for 10 min. Cell pellets were suspended in 1 ml Z buffer (40 mM NaH₂PO₄, 60 mM Na₂HPO₄, 1 mM MgSO₄, 10 mM KCl, and 38 mM β-mercaptoethanol) supplemented with 200 μg ml⁻¹ lysozyme. Resuspensions were incubated at 37 °C for 15 min. Reactions were started by adding 200 μl of 4 mg ml⁻¹ ONPG (2-nitrophenyl-β-D-galactopyranoside) and stopped by adding 500 μl of 1 M Na₂CO₃. Samples were then briefly centrifuged down at 5000 rpm for 1 min. The soluble fractions were transferred to cuvettes (VWR), and absorbance of the samples at 420 nm was recorded using a Bio-Rad spectrophotometer. The β-galactosidase-specific activity was calculated according to the equation (Abs₄₂₀/time × OD₆₀₀) × dilution factor × 1000. Assays were conducted in triplicate.

**Iron quantification by ICP-MS.** Quantification of matrix and intracellular iron followed a published protocol with minor modifications[10]. For the quantification of matrix-associated iron, the colony or pellicle biofilms after incubation at 30 °C for the indicated time, were collected and disrupted in 4 ml of 1× chelex-treated PBS buffer with pipetting. Samples were then mildly sonicated with 5-s pulse at the 1.5 output scale for three times (Branson, Model W185), normalized to OD₆₀₀ = 1.0 by using 1× chelex-treated PBS buffer. One milliliter of normalized samples was centrifuged at 14,000 rpm for 1 min, and the supernatant was ready for quantification of the matrix-associated iron. For measurement of the intercellular iron concentration, cell pellets after centrifugation were washed once with buffer 1 (1× PBS buffer, 0.1 M EDTA) and then twice with buffer 2 (1× chelex-treated PBS

buffer). Cell pellets were resuspended in 400 μl of buffer 3 (1× chelex-treated PBS buffer, 75 mM $NaN_3$, 1% Triton X-100) and incubated at 37 °C for 90 min. Lysed cells were centrifuged and the total protein content was quantified using a Bradford assay. Samples were then mixed with 600 μl buffer 4 (5% $HNO_3$, 0.1% (v/v) Triton X-100) and heated in a 95 °C sand bath for 30 min. The debris was removed by centrifugation. The total iron in all samples was analyzed by Bruker Aurora M90 ICP-MS. The $OD_{600}$ values of the biofilm samples were later converted to colony-forming units using standard plating techniques. The total iron concentration was presented as molar per cell (mean ± SD; $n = 10$).

**Electrochemical characterization.** To characterize electron transfer between the *B. subtilis* biofilm and ferric ions in electrolyte, CV was performed in a three-electrode cell by an electrochemical workstation (BioLogic). An RVC foam (3 cm × 3 cm × 6.5 mm) was used as the working electrode, while a platinum plate (1 cm × 1 cm) and Ag/AgCl (saturated, KCl) were used as the counter and reference electrode, respectively. The MSgg medium, which contained 50 μM of $FeCl_3$ as the unique electron-acceptable component in this system, was directly used as electrolyte. The pellicle biofilm of *B. subtilis* was coated on the surface of RVC foam electrode prior to the measurement. CV tests were operated in the range of −1.0~ +1.0 V at a scan rate of 20 mV s$^{-1}$.

**Oxygen profiling of biofilms.** For oxygen profiling, colony biofilms of *B. subtilis* 3610 were developed as described above. A Unisense Microprofiling System with a 50-μm-tip Clark-type oxygen microsensor (Unisense OX-50) was used to measure the oxygen concentrations in the colony biofilm. The oxygen microsensor was calibrated according to the manufacturer's instructions, and measurements were taken throughout the depth of the biofilm. During the measurement, the step size was 10 μm, the measurement time period was 5 s, and the wait time between different measurements was 5 s. Three different colonies were tested, and representative data were shown.

**Redox profiling of biofilms.** For redox profiling, 3610 cells were grown to exponential phase in LB broth at 37 °C with shaking. One hundred microliters of the culture was then inoculated into 100 ml of MSgg liquid medium in a 200 ml beaker and incubated at 30 °C for 3 days for pellicle biofilm development. For the assay, a 500-μm-tip redox microelectrode with a reference microelectrode (Unisense RD-500 and REF-RM) was used to measure the extracellular redox potential in the pellicle biofilm by using the same workstation used in the oxygen profiling. After calibrating the redox micro-electrode according to the manufacturer's instructions, redox measurements were taken throughout the depth of the pellicle biofilm. During the measurement, the step size was 50 μm, the measurement time period was 5 s, and the wait time between measurements was 5 s. The redox potential was set to zero at the surface of the pellicle biofilm, and relative values within the pellicle biofilm were plotted. Three different pellicle biofilms were used for measurement and representative data were shown.

**Measurement of membrane potential.** To measure the membrane potential of the cells in the biofilm, MSgg agar plates supplemented with 10 μM ThT, the fluorescent dye for membrane potential measurement was used. ThT is a cationic dye that acts as a Nernstian voltage indicator, which accumulates in cells as membrane potential decreases (accumulation thus negatively correlates to cell membrane potential) and the fluorescence intensity in the cells therefore increases[39].

**Quantitative real-time PCR.** Total RNAs of the cells in different treatment groups were extracted by using TRIzol (Invitrogen) according to the manufacturer's instructions. Isolated RNAs were reverse transcribed into single-stranded complementary DNA (cDNA) using a High Capacity cDNA Reverse Transcription Kit (Applied Biosystems). Primers used for RT-qPCR are listed in Supplementary Table 2. qRT-PCR was performed by using Fast SYBR$^{TM}$ Green Master Mix (Applied Biosystems) and a Step One Plus Real-Time PCR system (Applied Biosystems). The 16S rRNA gene was used as an internal reference. The relative expression of specific genes was calculated by using the $2^{-\Delta\Delta CT}$ method[36].

**Reporting summary.** Further information on research design is available in the Nature Research Reporting Summary linked to this article.

## Data availability

The source data underlying Fig. 2b, b, Fig. 3e, Fig. 4d, e, Fig. 5a–d, Fig. 6b, and Supplementary Figs. 1 and Fig. 2b are provided as a Source Data file.

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

## Acknowledgements

We thank Prof. Wei Qiao and Dr. Dongmin Yin from China Agricultural University (Beijing, China) for the help with redox potential measurement, Prof. Kehui Cui and Dr. Guo Zhang from Huazhong Agricultural University (Hubei, China) for the help of oxygen profiling, and Prof. Akram Alshawabkeh and Dr. Long Chen at Northeastern University (Boston, MA, USA) for providing technical help of the work. This work was supported by a grant from the National Science Foundation (US) to Y.C. (MCB1651732), and a grant from the National Natural Science Foundation of China (NSFC, No. 31671831) and Special Fund for Agro-scientific Research in the Public Interest of China (201303014-4) to P.L. Y.Q. was supported by a scholarship from the China Scholarship Council (file no. 201606350112). ICP-MS measurements were financially supported by a grant from the National Science Foundation (US, CBET-1254245).

## Author contributions

Y.Q., Y.H. and Q.S. performed the experiments. P.L.-C. provided substantial technical assistance. Y.Q., P.L. and Y.C. designed the experiments. Y.Q. and Y.C. wrote the manuscript.
