## [Peer Review File · Nature Communications]

Reviewers' comments:

Reviewer #1 (Remarks to the Author):

This article by Qin et al. looks at the interplay between iron and biofilm formation in *Bacillus subtilis*. In particular, the authors suggest an important role for the siderophore precursor DHB for iron during biofilm formation, and examine why an extremely extracellular high iron concentration is required for biofilm formation. They suggest that iron complexed to DHB in the biofilm would favour external electron transfer for cell no able to access oxygen, and this process would be essential for maintaining a strong membrane potential of cells required for matrix production. This manuscript contains a mix of great observations, and some less supported observations. Key observations need to be strengthened, and others could be excluded from the manuscript to focus on the main story, which is extremely interesting and novel. The manuscript in general also needs to be tightened up.

Major comments

I have concerns about two main points of this article.

Adaptative production of DHB over bacillibactin in biofilms:

The argument made by the author for the advantage of producing DHB over bacillibactin is neat, and the qPCR presented in 2C is interesting. However, qPCR was done on cells grown shaking condition, which cannot be compared to standing/biofilm formation. Accessibility to iron (diffusion of the ions and of siderophores), and the effect of the matrix on the cells will be different between these two conditions. Consequently, results obtained with cells in shaking conditions cannot be transferred to biofilm conditions. Of note, quantification of DHB and bacillibactin during pellicle formation in MSgg was recently published (Rizzi et al 2019) and indeed, DHB is produced at a higher level than bacillibactin, but only 10 fold. The authors could refer to these data, and perform qPCR on biofilms to confirm their experimental system; or for a stronger point quantify both molecules in biofilm and in spent media. The manuscript in its current form does not demonstrate that DHB is produced and/or present at much higher quantity than bacillibactin in biofilms.

Heterogeneity of respiratory electron transfer:

The heterogeneity in respiratory electron transfer is never demonstrated, but inferred from several experiments, some of which are on colonies, others on pellicles. While the demonstration is logical, a high-impact publication requires a more clean proof of this heterogeneity, using one or the other type of biofilm to be consistent. Authors need to demonstrate the functionality of electron transfer chain in cells in the bottom of the biofilm for pellicle (for example, with thioflavin T – reporter for EET enzymes), in parallel with the decreasing redox potential (5D). Figure 4B is not sufficient for this demonstration, since it uses a colony (and not pellicle as in 5D) and we do not know if the cells presented are “top” or “bottom” cells (or both).

Minor comments

Figure 2A and section line 137 – 159 are extremely speculative. The nature of the brown pigment is unknown, and could be related to iron solubilisation – but also could be just a secondary metabolite linked to biofilm formation. Production of the pigment is loss in *dhbA* mutant, and restored by DHB addition, but so is biofilm formation. This result does not prove anything.

Line 48. Introduction

Line 77. What do you mean by “bacillibactin-like”? Catechol siderophore? Most pathogens need siderophore to survive in certain environmental conditions.

Line 87. delete "to be added to the biofilm-inducing media iron "

Line 102. Non-ribosomal peptide

Line 123, 124. dihydrobenzoate

Line 187-197 belong to the discussion.

Line 242-253 are speculative and not useful to the manuscript.

Line 257-309 : maybe focussing on the mutants shown in 3C would tightened the message?

Line 327 and others: I am unsure of the terms "morphological robustness of the cells". There is an obvious morphological difference in the outer and inner colony, but is it due to the morphology of the cells , or to better growth/matrix production?

I suggest taking line 367-371 and Figure 4E out of the manuscript, they do not strengthen it.

Figure 5A : The level of intracellular iron should be presented per cell, and not per OD600, in order to be compare with the current literature.

Line 454-457 should be part of the discussion.

Discussion is very long, and part of it repeats the result section. The section needs to be re-visited extensively.

Line 520: evolutionary perspective?

Line 531: may not be an issue

Line 558- 573 are much too speculative.

Reviewer #2 (Remarks to the Author):

The major novel claims made in this manuscript, by Qin et al, are:

- 1) The *dhb* operon is induced upon exposure to increased iron in a *AbrB* dependent manner, resulting in the production of a bacillibactin precursor that is essential for biofilm formation under these conditions. I believe that that a functional role for this siderophore precursor is a novel and interesting finding.
- 2) The DHB precursor solubilizes ferric iron in the biofilm, based on the loss of a brown pigment (presumably iron) around colony biofilms in the *dhbA* mutant.
- 3) Electron transport chain genes are induced in the high iron biofilm growth conditions. Three of these (*sdhC*, *gpsA*, and *menF*), when deleted, were attenuated for biofilm but not planktonic growth, and chemically blocking the ETC had a similar effect.
- 4) The *sdhC* mutant has a compromised membrane potential and displays altered biofilm matrix production.
- 5) In these growth conditions, iron concentrates in the matrix of the outer part of the colony biofilm, in the same areas where oxygen concentrations decrease in the deeper layers of the biofilm. The biofilm is also electroactive as assessed by CV assay. Therefore matrix-associated iron may serves as a terminal electron acceptor during EET in , leading to ferric iron reduction.

While there are some interesting aspects of this manuscript, I do not think the data support all of

the claims made. With specific reference to the claims above:

2) This claim would be better supported with data showing that ferric iron is indeed bound by DHB under these biofilm growth conditions, and is solubilized in the WT but not *dhbA* mutant. Please also show the chemical complementation experiments referenced in line 146 and 156.

3) It is difficult to assess the images in Fig 4A. I would like to see quantification of this data, and inclusion of a positive control such as a membrane depolarizing agent. A link to glutamate is suggested here, and it is inferred that this may therefore suggest "electronic signalling", but this was superficial and based on one colony biofilm growth experiment. Since the model is that DHB solubilizes this matrix-associated

5) Since EET is not promoting iron uptake in the biofilm, and since iron does not appear to directly promote bacterial growth, I **think** the authors are suggesting that iron should be promoting ATP production to enhance biofilm growth. Has this been tested? A final note, the model (Fig 6) suggests that the metabolism demonstrated here promotes iron uptake into the cells, but there is no data supporting that (no difference in intracellular iron in biofilm cells in Fig 5B), since Fig 5A is showing exponentially growing cells. Related, is a *dhbA* or *sdhC* mutant attenuated for EET, as would be predicted?

Minor comments:

1) The writing style is somewhat colloquial (eg line 153, I doubt bacteria are "wise"), sometimes imprecise, and with increasing typos as the manuscript goes on. These things can be addressed at the editorial level, but nonetheless distract from the reading of the manuscript.

2) Fig 5B. Clarify which box is the outer and central area. It is inferred, but should be made explicit.

Reviewer #3 (Remarks to the Author):

Heterogeneity in Respiratory Electron Transfer and Adaptive Iron Utilization in a Bacterial Biofilm

In their manuscript Qin et al. describe the role of the siderophore precursor 2,3-DHB in iron uptake and the connection of iron availability to biofilm formation. The major claims are that *B. subtilis* is able to adapt to different iron levels in the medium by secretion of siderophore compounds DHB and bacillibactin, but in a concentration dependent manner. At higher iron concentrations the DHB instead of the full bacillibactin is secreted. DHB has a lower affinity, but also a lower metabolic cost. Iron is found at high levels in the extracellular matrix. This is novel and well executed. Certainly an interesting paper in the field that will broaden our understanding of biofilm formation in *B. subtilis*.

The paper then provides a range of hypothesis for mechanisms, which seem to be premature and require more experiments or should be removed or presented less prominently. For instance:

The authors claim that the biofilms are highly conductive. This is not clearly demonstrated. Page 16 line 449: The cyclic voltammogram presented in figure 5E shows oxidative and reductive waves that can be attributed to iron oxidation/reduction, respectively. However, this does not at all prove that the biofilm is responsible for these redox peaks. At least a blank CV with full medium including the DHB would be needed as a comparison. It is quite likely that the electrode will electrochemically reduce and oxidise the complexed metals. Also this will not prove that the biofilm is conductive but rather show that there is a redox reaction happening at the electrode/biofilm interface (the surface layer of the electrode). Any conclusion in the paper resting on this information needs to be revisited, or additional experimental data (showing no peaks without cells) has to be added. The authors could consider to conduct confocal RAMAN spectroscopy to study the redox state of cytochromes throughout the biofilm. This may give a hint on redox state within the biofilm. Figure 5 F shows a sketch of a bioelectrochemical system used for CV measurement. The picture is wrong. The counter electrode CE is missing (or actually it seems the CE is in fact named as the reference electrode RE, which is missing.)

Figure 5 B,C : How thick is the biofilm? Could the Oxygen measurement be done in the agar at 250 μ M penetration? If so the O₂ concentration would be very similar in all regions of the colony.
Fig 5B: How was the data normalized to OD, when cells are in a biofilm matrix?

It is quite common to observe anodic respiration of metal complexes by obligate aerobes (<https://doi.org/10.1186/s13068-016-0452-y>), even without growth and both for gram positive and gram negative strains such as *C.glutamicum* und *P.putida*. To conclude from the presented data that the *B.subtilis* in the lower layers is (Page 18 Line 534 following) using IET for growth is pure speculation. It is possible that those cells use IET to transfer electrons, but if this is sufficient to enable growth can't be demonstrated. In fact, how do the authors know that those cells grow at all? The observed membrane potential does only give a distribution of protons, but no information on proton fluxes and ATP generation rates? In the cited work above, *P.putida* also exhibited elevated ATP yields using iron respiration while the cells were dying...

Page 12 line 353 The role of glutamate in biofilm formation: how relevant is the finding that glutamate in the artificial environment of an agar plate plays a role in biofilm formation? Is it to be expected to play a role in the natural habitat? Could it just be the case that the DHB synthesis relies on Glutamate? At least in bacillibactin biosynthesis the amino donor for the amino acids Serine (glycine) and threonine is glutamate (EC 2.6.1.52 ; EC 2.6.1.1) and thus lower glutamate concentrations intracellularly might suppress the pathway? P20L572: What about uptake of DHB complexed with iron? Do you need a membrane potential for this?

Page 17 line 508: The reference 51 deals with catecholamines as far as I can tell. The sensitivity towards catechol of the named microbes only occurs at concentrations of 5 mM and above according to the cited ref 52. If this has an evolutionary relevance is hard to tell. Do plants secrete that high amounts of catechol? To me this section is too speculative.

Other comments

The paper is generally understandable, but some sections seem to be of different quality. Careful language polishing (see below) and attention to editorial details are necessary.

Results section and discussion section: The results section is not a pure results section. In fact it reads more like a Results and Discussion section, yet another discussion section is added. This is a bit confusing and should be more clearly separated or lumped altogether.

This is a question of style but maybe consider using the personal pronoun less frequently in the results section. The presented results should generally be achieved by everybody repeating the work not just YOU. Using personal pronoun all the time makes it sound less objective to me.

Page 5 Line 123 & 124: Check spelling of 2,3- DHB.

Page 6 line 148 and following: Check spelling of Bacillibactin in comparison to the rest of the manuscript

Page 6 Line 145: Is DHB really colourless? In my experience 3,4 DHB, an isomer of 2,3DHB used as chelator in minimal media for Corynebacteria has a purple colour once complexed with iron.

Page 6 line 149: There is also a whole body of work using 3,4DHB as iron chelator in minimal media of *Corynebacterium glutamicum*. Page 17 Line 506: DHB does not cause a growth defect in *Corynebacterium glutamicum* at concentrations needed for iron chelation...

Page 7 Line 187: Check sentence

Caption of Figures 2,3,4 References indicated but not added!

Page 18 Line 520: Check: Revolutionary perspective

Reviewer #1 (Remarks to the Author):

This article by Qin et al. looks at the interplay between iron and biofilm formation in *Bacillus subtilis*. In particular, the authors suggest an important role for the siderophore precursor DHB for iron during biofilm formation, and examine why an extremely extracellular high iron concentration is required for biofilm formation. They suggest that iron complexed to DHB in the biofilm would favour external electron transfer for cell no able to access oxygen, and this process would be essential for maintaining a strong membrane potential of cells required for matrix production. This manuscript contains a mix of great observations, and some less supported observations. Key observations need to be strengthened, and others could be excluded from the manuscript to focus on the main story, which is extremely interesting and novel. The manuscript in general also needs to be tightened up.

- We thank the reviewer's comments. We hope that we have addressed all the concerns raised by the reviewer in the revised manuscript. We also took out some of the less relevant results and substantially rewrote the entire manuscript as suggested. The revised manuscript is more focused on the more important findings.

Major comments

I have concerns about two main points of this article.

Adaptative production of DHB over bacillibactin in biofilms:

The argument made by the author for the advantage of producing DHB over bacillibactin is neat, and the qPCR presented in 2C is interesting. However, qPCR was done on cells grown shaking condition, which cannot be compared to standing/biofilm formation. Accessibility to iron (diffusion of the ions and of siderophores), and the effect of the matrix on the cells will be different between these two conditions. Consequently, results obtained with cells in shaking conditions cannot be transferred to biofilm conditions. Of note, quantification of DHB and bacillibactin during pellicle formation in MSgg was recently published (Rizzi et al 2019) and indeed, DHB is produced at a higher level than bacillibactin, but only 10 fold. The authors could refer to these data, and perform qPCR on biofilms to confirm their experimental system; or for a stronger point quantify both molecules in biofilm and in spent media. The manuscript in its current form does not demonstrate that DHB is produced and/or present at much higher quantity than bacillibactin in biofilms.

- We completely agree with the reviewer. In the revised manuscript, we repeated the qPCR experiment using biofilm samples (**Figs. 2B and 3E in the revised manuscript**). We also referred to the recent publication by Rizzi et al 2019 for quantification of DHB and bacillibactin in biofilm samples and added the citation in the revised text (lines 148-150).

Heterogeneity of respiratory electron transfer:

The heterogeneity in respiratory electron transfer is never demonstrated, but inferred from several experiments, some of which are on colonies, others on pellicles. While the demonstration is logical, a high-impact publication requires a more clean proof of this heterogeneity, using one

or the other type of biofilm to be consistent. Authors need to demonstrate the functionality of electron transfer chain in cells in the bottom of the biofilm for pellicle (for example, with thioflavin T – reporter for EET enzymes), in parallel with the decreasing redox potential (5D). Figure 4B is not sufficient for this demonstration, since it uses a colony (and not pellicle as in 5D) and we do not know if the cells presented are “top” or “bottom” cells (or both).

- As suggested, we modified a technique used in our previous study (Beauregard et al. PNAS 2013) that allowed us to separate top and bottom cells of a pellicle biofilm by using a metal mesh. We then demonstrated the functionality of the ETC in the bottom and top cells of the pellicle biofilm by using the ThT dye. This new result was provided as **Figs. 5E-F in the revised manuscript and the results were discussed in lines 366-376**. Our observations from the new experiment show that cells at the bottom of the pellicle biofilm demonstrate meaningful robustness in membrane potential, although not as strong as the cells at the top.

Minor comments

Figure 2A and section line 137 – 159 are extremely speculative. The nature of the brown pigment is unknown, and could be related to iron solubilisation – but also could be just a secondary metabolite linked to biofilm formation. Production of the pigment is lost in *dhbA* mutant, and restored by DHB addition, but so is biofilm formation. This result does not prove anything.

- We agree. We took out this section about the loss of pigment in the *dhbA* mutant in the revised manuscript.

Line 48. Introduction

- We fixed the error (line 45).

Line 77. What do you mean by “bacillibactin-like”? Catechol siderophore? Most pathogens need siderophore to survive in certain environmental conditions.

- We changed the description to “In most pathogens, siderophores are essential for the bacteria to survive in the host or environment”. (lines 74-75).

Line 87. delete “to be added to the biofilm-inducing media iron “

- We deleted these words as suggested.

Line 102. Non-ribosomal peptide

- We changed the words accordingly (line 98).

Line 123, 124. dihydrobenzoate

- We fixed the errors (lines 120-121).

Line 187-197 belong to the discussion.

- We moved the section to the discussion (lines 472-476)

Line 242-253 are speculative and not useful to the manuscript.

- We deleted this section in the revised manuscript as the reviewer suggested.

Line 257-309 : maybe focusing on the mutants shown in 3C would tightened the message?

- We thank the reviewer for the suggestion. We took out those the less relevant results and discussion in this section, and focused on the mutants in Fig. 3C which show the biofilm phenotypes (lines 210-216).

Line 327 and others: I am unsure of the terms “morphological robustness of the cells”. There is an obvious morphological difference in the outer and inner colony, but is it due to the morphology of the cells , or to better growth/matrix production?

- We deleted the words of “morphological robustness of the cells” and rewrote this section (lines 287-290).

I suggest taking line 367-371 and Figure 4E out of the manuscript, they do not strengthen it.

- As the reviewer suggested, we took out Fig. 4E and the corresponding text in the revised manuscript.

Figure 5A : The level of intracellular iron should be presented per cell, and not per OD600, in order to be compare with the current literature.

- As suggested, we did a control experiment that allowed us to convert OD600 values of the biofilm samples to the number of cells (CFU) per ml using standard plating method. In the revised manuscript, the iron concentrations were presented as molar per cell in **Figs 5A-B**.

Line 454-457 should be part of the discussion.

- We moved the text in this section to the discussion as suggested (lines 467-472).

Discussion is very long, and part of it repeats the result section. The section needs to be re-visited extensively.

- We thank the reviewer for the suggestion. We rewrote the discussion. We hope in the revised manuscript, discussion is significantly shortened and streamlined.

Line 520: evolutionary perspective?

- We fixed the error.

Line 531: may not be an issue

- We fixed the error.

Line 558- 573 are much too speculative.

- Increasing evidence from an array of recent studies (Prindle et al. 2015 Nature; Humphries et al, 2017 Cell; Liu et al. 2017, Science, etc) suggest that membrane potential plays a uniquely important in cell-cell communication and multicellular development in bacteria. We understand that the mechanistic details are still lacking, but we would really like to keep this discussion about the potential importance of membrane potential in *B. subtilis* biofilm development, which may allow additional thinking beyond the presented data in the study from the readers.

We did rewrite this section and tried to remove sentences that are too speculative (lines 435-447).

Reviewer #2 (Remarks to the Author):

The major novel claims made in this manuscript, by Qin et al, are:

- 1) The *dhb* operon is induced upon exposure to increased iron in a *AbrB* dependent manner, resulting in the production of a bacillibactin precursor that is essential for biofilm formation under these conditions. I believe that that a functional role for this siderophore precursor is a novel and interesting finding.
- 2) The DHB precursor solubilizes ferric iron in the biofilm, based on the loss of a brown pigment (presumably iron) around colony biofilms in the *dhbA* mutant.
- 3) Electron transport chain genes are induced in the high iron biofilm growth conditions. Three of these (*sdhC*, *gpsA*, and *menF*), when deleted, were attenuated for biofilm but not planktonic growth, and chemically blocking the ETC had a similar effect.
- 4) The *sdhC* mutant has a compromised membrane potential and displays altered biofilm matrix production.
- 5) In these growth conditions, iron concentrates in the matrix of the outer part of the colony biofilm, in the same areas where oxygen concentrations decrease in the deeper layers of the biofilm. The biofilm is also electroactive as assessed by CV assay. Therefore matrix-associated iron may serve as a terminal electron acceptor during EET in , leading to ferric iron reduction.

While there are some interesting aspects of this manuscript, I do not think the data support all of the claims made. With specific reference to the claims above:

2) This claim would be better supported with data showing that ferric iron is indeed 1) bound by DHB under these biofilm growth conditions, and is 2) solubilized in the WT but not *dhbA* mutant. Please also show the 3) chemical complementation experiments referenced in line 146 and 156.

- Based on other reviewers' comments, we have removed the results about loss of pigment in the *dhbA* mutant and complementation of the pigment by adding pure DHB from the revised manuscript. The chemical complementation experiment remains in the revised manuscript, but only focused on rescue of the biofilm phenotype (**Fig. 1D**), not pigment recovery.

3) It is difficult to assess the images in Fig 4A. I would like to see quantification of this data, and inclusion of a positive control such as a membrane depolarizing agent. A link to glutamate is suggested here, and it is inferred that this may therefore suggest "electronic signalling", but this was superficial and based on one colony biofilm growth experiment. Since the model is that DHB solubilizes this matrix-associated

- We quantified the results in Fig. 4A using Image J. The quantitative analyses results are now provided in **Figs. 4D-E in the revised manuscript**.

- In addition, we added a control by using a membrane depolarizing agent CCCP. The new results are shown in **Fig. 4C in the revised manuscript**, and quantification in Figs. 4D-E

- About glutamate uptake, we agree the reviewer that it is superficial. Therefore, as suggested, we took out Fig 4E and the related result description. We only very briefly discussed the

possible impact of membrane potential on glutamate uptake and essential roles of glutamate in biofilm development in the discussion section in the revised manuscript (lines 443-447).

5) Since EET is not promoting iron uptake in the biofilm, and since iron does not appear to directly promote bacterial growth, I *think* the authors are suggesting that iron should be promoting ATP production to enhance biofilm growth. Has this been tested? A final note, the model (Fig 6) suggests that the metabolism demonstrated here promotes iron uptake into the cells, but there is no data supporting that (no difference in intracellular iron in biofilm cells in Fig 5B), since Fig 5A is showing exponentially growing cells. Related, is a *dhbA* or *sdhC* mutant attenuated for EET, as would be predicted?

- We claim in the manuscript that iron may promote strong membrane potential through EET via two different mechanisms (upregulation of EET enzymes and extracellular electron transfer). Membrane potential may have important roles in biofilm development such as in substrate transport and ion signaling, in addition to, but not necessarily in, promoting ATP production (obviously an intuitive thought). Our hypothesis has not been on ATP production yet, we therefore did not measure if iron promotes ATP production. In the discussion, we briefly speculated how membrane potential may promote biofilm formation (lines 435-447).

We agree that the difference in iron uptake between planktonic cells and biofilm cells is mild (Figs. 5A-B), in part could be due to iron homeostasis. We do think the difference could be meaningful in influencing iron-involved processes.

And both the *dhbA* and *sdhC* mutants have lowered membrane potential. For *sdhC*, this is supported by the evidence shown in **Fig. 4A and 4B**. the *dhbA* mutant is quite similar in membrane potential status (Qin, unpublished). We think this is likely due to attenuated EET (Δ *sdhC* results in defective EET system while Δ *dhbA* results in lack of iron uptake, less active EET, and lack of soluble iron for extracellular electron transfer)

Minor comments:

1) The writing style is somewhat colloquial (eg line 153, I doubt bacteria are “wise”), sometimes imprecise, and with increasing typos as the manuscript goes on. These things can be addressed at the editorial level, but nonetheless distract from the reading of the manuscript.

- We apologize for that. We rewrote the manuscript and thoroughly checked the grammar errors. We hope that the revised manuscript is much improved on writing.

2) Fig 5B. Clarify which box is the outer and central area. It is inferred, but should be made explicit.

- We added “outer” and “inner” labels on the top of the boxes and also used different colors for the boxes in Fig. 5B.

Reviewer #3 (Remarks to the Author):

Heterogeneity in Respiratory Electron Transfer and Adaptive Iron Utilization in a Bacterial Biofilm

In their manuscript Qin et al. describe the role of the siderophore precursor 2,3-DHB in iron uptake and the connection of iron availability to biofilm formation. The major claims are that *B. subtilis* is able to adapt to different iron levels in the medium by secretion of siderophore compounds DHB and bacillibactin, but in a concentration dependent manner. At higher iron concentrations the DHB instead of the full bacillibactin is secreted. DHB has a lower affinity, but also a lower metabolic cost. Iron is found at high levels in the extracellular matrix. This is novel and well executed. Certainly an interesting paper in the field that will broaden our understanding of biofilm formation in *B. subtilis*.

The paper then provides a range of hypothesis for mechanisms, which seem to be premature and require more experiments or should be removed or presented less prominently. For instance:

The authors claim that the biofilms are highly conductive. This is not clearly demonstrated. Page 16 line 449: The cyclic voltammogram presented in figure 5E shows oxidative and reductive waves that can be attributed to iron oxidation/reduction, respectively. However, this does not at all prove that the biofilm is responsible for these redox peaks. At least a blank CV with full medium including the DHB would be needed as a comparison. It is quite likely that the electrode will electrochemically reduce and oxidise the complexed metals. Also this will not prove that the biofilm is conductive but rather show that there is a redox reaction happening at the electrode biofilm interface (the surface layer of the electrode). Any conclusion in the paper resting on this information needs to be revisited, or additional experimental data has to be added. The authors could consider to conduct confocal RAMAN spectroscopy to study the redox state of cytochromes throughout the biofilm. This may give a hint on redox state within the biofilm. Figure 5 F shows a sketch of a bioelectrochemical system used for CV measurement. The picture is wrong. The counter electrode CE is missing (or actually it seems the CE is in fact named as the reference electrode RE, which is missing.)

- To address the points raised by the reviewer, we fixed the error in the sketch in **Fig. 6A** (Fig. 5F in the previously submitted manuscript). We also added a blank CV with medium only, and with medium and DHB as two control experiments. We also repeated the CV measurement with the 3610 biofilm. The new results are shown in **Fig. 6B in the revised manuscript**.

We also rewrote the results and figure legend to make sure that we are clear but also cautious in interpreting our results from the CV experiments.

Figure 5 B,C : How thick is the biofilm? Could the Oxygen measurement be done in the agar at 250 μ M penetration? If so the O₂ concentration would be very similar in all regions of the colony. Fig 5B: How was the data normalized to OD, when cells are in a biofilm matrix?

- The colony biofilm is about 360 μ m in depth. During the measurement in colony biofilm, the oxygen concentration decreased to almost zero at the depth about 200 μ m and began to mildly increase at the depth of 360 μ m when the microelectrode began to spur into the agar. The oxygen concentration is about 20~30 μ M in the agar, about 10-fold less than the oxygen concentration in the air (250 μ M).
- In the revised manuscript, we changed OD₆₀₀ to per cell based on one reviewer's suggestion. In the previous version of the manuscript, biofilm samples were collected from outer or inner region, resuspended, and measured for OD₆₀₀. The biofilm normally consists of 70% of the

cells and 30% of the extracellular matrix. We did standard plating experiment to convert OD600 values of biofilm samples to CFU. In the revised manuscript, we presented the iron concentration as molar per cell (**Fig. 5A-B**).

It is quite common to observe anodic respiration of metal complexes by obligate aerobes (<https://doi.org/10.1186/s13068-016-0452-y>), even without growth and both for gram positive and gram negative strains such as *C.glutamicum* und *P.putida*. To conclude from the presented data that the *B.subtilis* in the lower layers is (Page 18 Line 534 following) using IET for growth is pure speculation. It is possible that those cells use IET to transfer electrons, but if this is sufficient to enable growth can't be demonstrated. In fact, how do the authors know that those cells grow at all? The observed membrane potential does only give a distribution of protons, but no information on proton fluxes and ATP generation rates? In the cited work above, *P.putida* also exhibited elevated ATP yields using iron respiration while the cells were dying...

- We agree with reviewer, we tried to to be cautious here. On the other hand, our hypothesis indeed stopped at the electron transfer and establishment of membrane potential, which we have evidence presented in this study, but not further into ATP generation and cell growth. We believe that membrane potential may have important roles in biofilm development such as in substrate transport (e.g. charged glutamate which is essential for robust biofilm formation) and ion signaling, in addition to, but not necessary in, promoting ATP production and cell growth (clearly an intuitive thought).

Nevertheless, we think the reviewer raised an interesting and important point, whether cells in the bottom, show growth or are even viable. We did additional experiments in the revised manuscript to investigate membrane potential status separately in the top and bottom layers of cells in the biofilm. We also performed live/death staining for these two layers of cells to simply look at cell viability. New results about membrane potential in the top and bottom layers of cells are provided in **Figs 5E-F in the revised manuscript**. The live/dead staining results are provided as a supplemental file (**Fig. S3**). In summary, bottom layers of cells are largely viable and demonstrate relatively robust membrane potential.

Page 12 line 353 The role of glutamate in biofilm formation: how relevant is the finding that glutamate in the artificial environment of an agar plate plays a role in biofilm formation? Is it to be expected to play a role in the natural habitat? Could it just be the case that the DHB synthesis relies on Glutamate? At least in bacillibactin biosynthesis the amino donor for the amino acids Serine (glycine) and threonine is glutamate (EC 2.6.1.52 ; EC 2.6.1.1) and thus lower glutamate concentrations intracellularly might suppress the pathway?

- Glutamate is very important in promoting robust biofilm formation. Replacing glutamate with other nitrogen sources greatly reduced biofilm development of *B. subtilis*. But we don't know why. We thank the reviewer for providing possibly interesting insights on the importance of glutamate, and physiological relevance of this amino acid in the natural habitat that is worth testing in future studies.

Based comments from two other reviewers about the relevance of the glutamate result (Fig. 4E in the previously submitted manuscript), we decided to remove these results and the corresponding test in the revised manuscript.

P20L572: What about uptake of DHB complexed with iron? Do you need a membrane potential for this?

- It was suggested by previous studies that uptake of DHB complexed with iron was carried out by an ABC transporter, namely FeuABC-YusV, in *B. subtilis* (Olliger et al. 2016 J. Bac). We also have genetic evidence that the transporter mutant (Δ feuA) is completely deficient in biofilm formation in regular MSgg medium, similar to that of dhbA mutant (Qin, unpublished).

Page 17 line 508: The reference 51 deals with catecholamines as far as I can tell. The sensitivity towards catechol of the named microbes only occurs at concentrations of 5 mM and above according to the cited ref 52. If this has an evolutionary relevance is hard to tell. Do plants secrete that high amounts of catechol? To me this section is too speculative.

- We took out this section entirely from the discussion in the revised manuscript based on reviewers' comments.

Other comments

The paper is generally understandable, but some sections seem to be of different quality. Careful language polishing (see below) and attention to editorial details are necessary.

- Thanks for pointing that out. We thoroughly rewrote the manuscript and we hope that the revised manuscript is now much improved in writing.

Results section and discussion section: The results section is not a pure results section. In fact it reads more like a Results and Discussion section, yet another discussion section is added. This is a bit confusing and should be more clearly separated or lumped altogether.

- We rewrote the result section, moved content in a couple of places from result to discussion. Also, we significantly shortened the discussion section, made it much more focused and streamlined.

This is a question of style but maybe consider using the personal pronoun less frequently in the results section. The presented results should generally be achieved by everybody repeating the work not just YOU. Using personal pronoun all the time makes it sound less objective to me.

- We checked throughout the result section in the manuscript and tried our best to remove personal pronoun in places where we felt appropriate.

Page 5 Line 123 & 124: Check spelling of 2,3- DHB.

- We fixed the errors.

Page 6 line 148 and following: Check spelling of Bacillibactin in comparison to the rest of the manuscript.

- We fixed the error.

Page 6 Line 145: Is DHB really colourless? In my experience 3,4 DHB, an isomer of 2,3DHB used as chelator in minimal media for Corynebacteria has a purple colour once complexed with

iron.

- When we described that DHB is colorless in the text, what we were referring is that in the solution, pure 2,3-DHB (Sigma) without added ferric iron is colorless. Clearly, when binding to iron, the complex will develop red/purple-ish color as the reviewer similarly suggested for 3,4-DHB.

Page 6 line 149: There is also a whole body of work using 3,4DHB as iron chelator in minimal media of *Corynebacterium glutamicum*.

- Thanks for the information. We added this information and references in the revised text (lines 132-134).

Page 17 Line 506: DHB does not cause a growth defect in *Corynebacterium glutamicum* at concentrations needed for iron chelation...

- Based on the reviewers' comments, we removed this section from the discussion in the revised manuscript.

Page 7 Line 187: Check sentence

- We fixed that.

Caption of Figures 2,3,4 References indicated but not added!

- Sorry for the mistake. We added back the references in the places indicated.

Page 18 Line 520: Check: Revolutionary perspective

- we fixed the error.

REVIEWERS' COMMENTS:

Reviewer #1 (Remarks to the Author):

This revised version is an excellent manuscript; all my concerns were addressed. The findings are novel and interesting, the main conclusions are well supported, and the whole story is tighter than in the previous version.

I would just like to point out that in Fig. 5A and B, which shows intracellular iron is at 1×10^{-13} . This number is very high, and does not fit with current literature (concentration is between 10^{-16} to 10^{-18} mole per bacteria, for *E. coli* and *B. subtilis*). Original 5a and b showed numbers at around 100 nanomole of iron/ OD600, and OD600, in our hand, is approximately 1 to 5×10^8 cells, which would give a final number around closer to 10^{-16} mole of iron per cell. That is worth double-checking.

Reviewer #2 (Remarks to the Author):

Major

New data - Fig 5. Pellicle and microcolony biofilms are very different beasts. I do not agree with the authors' attempt to correlate findings of oxygen availability in colony biofilms with membrane potential in pellicle biofilms. The authors should measure these parameters in the same biofilm in order to draw the conclusions they would like to make.

Related, the authors draw correlations between iron abundance in the outer areas of the colony biofilm and deeper layers of the pellicle biofilm, data are not provided to support iron as an extracellular/matrix-associated electron acceptor in the deeper layers of the colony biofilm as modeled in Fig 6. These speculations are easily tested using imaging MS.

Minor

Line 199-200. Re ferric ppt in the media – what data supports this? What is the point of mentioning it here?

Line 352. What is meant by pace? This implies a measurement over time, which I do not believe is happening.

Line 373. What about the results in 5B are consistent with the indicated finding?

Reviewer #3 (Remarks to the Author):

The authors have addressed my concerns sufficiently. One minor change is still needed: line 120/121 2,3-dihydroxybenoate was changed twice for misspelling but is now still wrong. This is the correct spelling: 2,3-dihydroxybenzoate

REVIEWERS' COMMENTS:

Reviewer #1 (Remarks to the Author):

This revised version is an excellent manuscript; all my concerns were addressed. The findings are novel and interesting, the main conclusions are well supported, and the whole story is tighter than in the previous version.

I would just like to point out that in Fig. 5A and B, which shows intracellular iron is at 1×10^{-13} . This number is very high, and does not fit with current literature (concentration is between 10^{-16} to 10^{-18} mole per bacteria, for *E. coli* and *B. subtilis*). Original 5a and b showed numbers at around 100 nanomole of iron/ OD600, and OD600, in our hand, is approximately 1 to 5×10^8 cells, which would give a final number around closer to 10^{-16} mole of iron per cell. That is worth double-checking.

-We thank the reviewer for pointing this out. After revisiting the raw data, we recognized an error in calculating the CFU numbers per OD600. The OD600 of the biofilm cells still falls into the range of 10^8 cells, and the iron concentration is about 10^{-16} per biofilm cell. We fixed the errors in the text and the figure.

Reviewer #2 (Remarks to the Author):

Major

New data - Fig 5. Pellicle and microcolony biofilms are very different beasts. I do not agree with the authors' attempt to correlate findings of oxygen availability in colony biofilms with membrane potential in pellicle biofilms. The authors should measure these parameters in the same biofilm in order to draw the conclusions they would like to make.

Related, the authors draw correlations between iron abundance in the outer areas of the colony biofilm and deeper layers of the pellicle biofilm, data are not provided to support iron as an extracellular/matrix-associated electron acceptor in the deeper layers of the colony biofilm as modeled in Fig 6. These speculations are easily tested using imaging MS.

- We appreciate the reviewer's comments and largely agree with the comments, but would like to point out that we did use the same colony biofilm setting to measure both the oxygen availability (Fig. 5C) and membrane potentials (Fig. 4B).

There are several limitations of this study we would like to acknowledge. First, for redox potential measurement, the thickness of a colony biofilm did not allow us to use the redox probe we applied in this study; second, the measurement of redox needs aqueous environment to form a circuit between the redox probe and the reference probe and therefore only pellicle biofilm can satisfy this requirement; third, for intracellular and extracellular iron measurement, only colony biofilm was applied instead of both.

To further address the reviewer's concern, we included and emphasized in the results and discussion sections our cautious opinions about the results obtained from different biofilm settings (pellicle and colony biofilms) and the above limitation of this study (lines 475-477, 541-549).

Minor

Line 199-200. Re ferric ppt in the media – what data supports this? What is the point of mentioning it here?

- We deleted this sentence from the revised text.

Line 352. What is meant by pace? This implies a measurement over time, which I do not believe is happening.

- We changed the word pace to rate to indicate that it is a measurement against biofilm depth (line 438).

Line 373. What about the results in 5B are consistent with the indicated finding?

- To avoid confusion, we delete “which was consistent with earlier results in this study (Fig. 5B)” from the sentence.

Reviewer #3 (Remarks to the Author):

The authors have addressed my concerns sufficiently. One minor change is still needed:
line 120/121 2,3-dihydroxybenoate was changed twice for misspelling but is now still wrong. This is the correct spelling: 2,3-dihydroxybenzoate

- We fixed the mistakes.